



**The role of anthropogenic aerosols in the anomalous cooling**
**from 1960 to 1990 in the CMIP6 Earth System Models**
Jie Zhang[1], Kalli Furtado[2*], Steven T. Turnock[2], Jane P. Mulcahy[2], Laura J.Wilcox[3],
Ben B. Booth[2], David Sexton[2], Tongwen Wu[1], Fang Zhang[1], Qianxia Liu[1]
[1]Beijing Climate Center, China Meteorological Administration, Beijing, China,100081
[2]Met Office Hadley Centre, Exeter, UK, EX1 3PB
[3]National Centre for Atmospheric Science, Department of Meteorology, University of Reading,
Reading, UK
*Corresponding to*: Kalli Furtado (kalli.furtado@metoffice.gov.uk)
**Abstract** The Earth System Models (ESMs) that participated in the 6[th] Coupled
Model Intercomparison Project (CMIP6) tend to simulate excessive cooling in surface
air temperature (TAS) between 1960 and 1990. The anomalous cooling is pronounced
over the Northern Hemisphere (NH) midlatitudes, coinciding with the rapid growth of
anthropogenic sulfur dioxide ($SO_2$) emissions, the primary precursor of atmospheric
sulphate aerosols. Historical simulations with and without anthropogenic aerosol
emissions indicate that the anomalous cooling in the ESMs is partly due to
overestimating anthropogenic aerosols and aerosol-forcing sensitivity. Structural
uncertainties between ESMs that contribute to these two factors have a larger impact
on the anomalous cooling than internal variability. CMIP6 simulations can also help
us to quantify the relative contributions of aerosol-forcing-sensitivity by
aerosol-radiation interactions (ARI) and by aerosol-cloud interactions (ACI).
However, even when the aerosol-forcing-sensitivity is similar between ESMs, the
relative contributions of ARI and ACI may be substantially different. The ACI
accounts for 64 to 87% of the aerosol-forcing-sensitivity and is the main source of
differences between the ESMs. The ACI can be further decomposed into a
cloud-amount term (which depends linearly on cloud fraction) and a cloud-albedo
term (which is independent of cloud fraction, to the first order). The large
uncertainties of cloud-amount term are responsible for the aerosol-forcing-sensitivity
differences and further the anomalous cooling differences among ESMs. The metrics



used here therefore provide a simple way of assessing the physical mechanisms
contributing to anomalous twentieth century cooling in any given ESM, which may
benefit future model developments.

**1. Introduction**

Surface air temperature (TAS) variation is an essential indicator of climate
change, and reproducing the evolution of historical TAS is a crucial criterion for model
evaluation. However, the historical TAS anomaly simulated by the models in the 6[th]
Coupled Model Intercomparison Project (CMIP6) is on average colder than that
observed in the mid-twentieth century, whereas the CMIP5 models tracked the
instrumental TAS variation quite well (Flynn and Mauritsen, 2020). This is surprising
because the transient climate response in CMIP6 models is generally higher than in
CMIP5 models (e.g., Flynn and Mauritsen, 2020; Meehl et al., 2020).
As a result of anthropogenic emissions, atmospheric aerosol concentrations
increased along with rising greenhouse gases, but with greater decadal variability.
Aerosols increased rapidly in the mid-twentieth century, predominantly due to US and
European emissions. There has been little change in the global total emissions since
1980, but there has been a shift in emission source regions. European and US
emissions have declined following the introduction of clean air legislation, while
Asian emissions have risen due to economic development. Although greenhouse
warming was concluded to be the dominant forcing for long-term changes (e.g.,
Weart, 2008; Bindoff et al., 2013), multidecadal variability in TAS and the reduced
rate of warming in the mid-twentieth century in particular, has been attributed to
aerosol forcing (e.g. Wilcox et al., 2013). Ramanathan and Feng (2009) noted that the
aerosol cooling effect might have masked as much as 47% of the global warming by
greenhouse gases in the year 2005, with an uncertainty range of 20~80%. The aerosol
cooling effect is mainly attributed to the ability of sulphate particles to reflect
incoming solar radiation and modify the microphysical properties of clouds (e.g.,
Charlson et al., 1990; Mitchell et al., 1995; Lohmann and Feichter, 2005). The



increase in anthropogenic aerosols was also responsible for weakening the
hydrological cycle between the 1950s and the 1980s (Wu et al., 2013).
There have been efforts to study the anomalous mid-twentieth century cooling in
the CMIP6 models. Flynn and Mauritsen (2020) suggested that aerosol cooling is too
strong in many CMIP6 models because there is no apparent relationship between the
warming trends simulated by models and their transient climate responses (TCRs)
before the 1970s. The warming trend is larger than observed post-1970 in CMIP6
models, offsetting the pre-1970s cooling. Dittus et al. (2020) found that historical
simulations can better capture the observed historical record by reducing the aerosol
emissions in HadGEM3-GC3.1, demonstrating an overly strong aerosol cooling
effect. They showed that simulations with large anthropogenic aerosol emissions had
greater cooling trends between 1951 and 1980, which were significantly different to
the observed trend, while simulations with smaller aerosol forcing were more
consistent with observations.
In this study we characterize the mid-twentieth century excessive cooling in
CMIP6 ESMs. In order to quantify the role of aerosol processes in this anomalous
cooling, historical experiments with and without anthropogenic aerosol emissions are
employed. The remainder of the paper is organized as follows. Section 2 introduces the
models, data, and a quantitative method to separate the aerosol forcing components.
The major features of anomalous cooling in CMIP6 ESMs are examined in section 3.
Section 4 investigates the possible reasons for the anomalous cooling. The relative
importance of aerosol-radiation interactions and aerosol-cloud interactions is
quantified and discussed in section 5. Conclusion is given in Section 6.

**2.   Model, data, and method**
**2.1 CMIP6 ESMs**
CMIP6 includes an unprecedented number of models with representations of
aerosol-cloud interactions. Many also have interactive tropospheric chemistry and



aerosol schemes. Six such ESMs are employed in this study: BCC-ESM1 (Wu et al.,
2020; Zhang et al., 2021), EC-Earth-AerChem (Noije et al., 2020), GFDL-ESM4
(Dunne et al., 2020), MPI-ESM-1-2-HAM (Mauritsen et al., 2019), NorESM2-LM
(Seland et al., 2020), and UKESM1-0-LL (Sellar et al., 2019). The surface air
temperature simulated in corresponding models with lower-complexity are also
examined: BCC-CSM2-MR (Wu et al., 2019b), EC-Earth3 (Döscher et al., 2021), and
MPI-ESM1-2-LR (Mauritsen et al., 2019) with prescribed tropospheric chemistry and
aerosol; GFDL-CM4 (Held et al., 2019), NorCPM1 (Bethke et al., 2019), and
HadGEM3-GC31-LL (Williams et al., 2017) with prescribed tropospheric chemistry
and interactive aerosol scheme. BCC-CSM2-MR, EC-Earth3, and MPI-ESM1-2-LR
prescribe the anthropogenic aerosol forcings using the MACv2-SP parameterization
(Stevens et al., 2017). MACv2-SP approximates the observationally constrained spatial
distributions of the monthly mean anthropogenic aerosol optical properties and an
associated Twomey effect. Except for BCC models, the horizontal resolutions of the
ESMs are the same as the corresponding lower-complexity models. A brief summary of
the ESMs and the lower-complexity models is introduced in Table 1.





**Table 1.** Information of the ESMs with interactive chemistry and aerosol scheme, as

well as the corresponding lower-complexity models.

| Modeling group | ESM (Atmospheric Resolution) | Lower-complexity models (Atmospheric Resolution) | Prescribed tropospheric chemistry | Prescribed aerosol | Number of members | References |
|---|---|---|---|---|---|---|
| Beijing Climate Center (BCC) | **BCC-ESM1:** the BCC Earth System Model version 1 (T42, 26 layers to 2.914 hPa) | **BCC-CSM2-MR:** the median resolution BCC Climate System Model version 2 (T106, 46 layers to 1.459 hPa) | Y | Y | 3 | Wu et al.(2019b, 2020); Zhang et al. (2021) |
| European consortium of meteorological services, research institutes, and high-performance computing centres | **EC-Earth-AerChem:** the EC-Earth configuration with interactive aerosols and atmospheric chemistry (T255, 91 layers to 0.01 hPa) | **EC-Earth3:** the EC-Earth version 3 (T255, 91 layers to 0.01 hPa) | Y | Y | 1 | Noije et al. (2020); Döscher et al. (2021) |
| US Department of Commerce/NOAA / Geophysical Fluid Dynamics Laboratory (GFDL) | **GFDL-ESM4:** the GFDL Earth System Model version 4 (C96, 49 layers to 1 hPa) | **GFDL-CM4:** the GFDL Climate Model version 4 (C96, 33 layers to 1 hPa) | Y | N | 1 | Dunne et al. (2020); Held et al. (2019) |
| Max Planck Institute for Meteorology (MPI) | **MPI-ESM-1-2-HAM:** the HAMMOZ-Consortium of MPI Earth System Model (T63, 47 layers to 0.01 hPa) | **MPI-ESM1-2-LR:** the lower-resolution version of MPI Earth System Model (T63, 47 layers to 0.01 hPa) | Y | Y | 3 | Mauritsen et al. (2019); |
| Norwegian Climate Center (NCC) | **NorESM2-LM:** the lower-resolution of Norwegian ESM version 2 (About 2º, 32 layers to 2 hPa) | **NorCPM1:** the Norwegian Climate Prediction Model version 1 (About 2º, 26 layers to 3 hPa) | Y | N | 3 | Seland et al. (2020); Bethke et al. (2019) |
| Met Office's Hadley Centre for Climate Prediction and Research (MOHC) | **UKESM1-0-LL:** U.K. Earth System Model version 1 (N96, 85 layers to 85 km) | **HadGEM3-GC31-LL:** the Hadley Centre Global Environment Model in the Global Coupled configuration 3.1 (N96, 85 layers to 85km) | Y | N | 3 | Sellar et al. (2019); Williams et al. (2017) |





**2.2 Data**

**Table 2** Variables used in this study.

| Variable name | CMIP6 diagnostic label | Description | Units |
|---|---|---|---|
| TAS | tas | Surface air temperature | $^{o}C$ |
| OSR | rsut | All-sky outgoing shortwave radiation at the top of atmosphere (TOA) | $W\ m^{-2}$ |
| OSRclr | rsutcs | OSR assuming clear sky | $W\ m^{-2}$ |
| mmrso4 | mmrso4 | Mass mixing ratio of sulphate aerosol in the atmosphere | kg kg-1 |
| CLT | clt | Total cloud amount | % |
| $r_{eff}$ | reffclwtop | cloud-top effective droplet radius | µm |
| loadSO4 | | Sulphate loading in the atmosphere, calculated by mmrso4 | mg $m^{-2}$ |
| OSRclr_hist | | Mean OSRclr in the historical simulation from 1850 to 1990 | $W\ m^{-2}$ |
| CLT_hist | | Mean CLT in the historical simulation from 1850 to 1990 | % |


The CMIP6 historical experiment and hist-piAer experiment are employed. The

historical experiment is forced by time-evolving, externally imposed natural and
anthropogenic forcings, such as solar variability, volcanic aerosols, greenhouse gases,
and aerosol emissions (Eyring et al., 2016). The hist-piAer experiment is designed by
the    CMIP6-endorsed    Aerosol    Chemistry    Model    Intercomparison    Project
(AerChemMIP; Collins et al., 2017). It is run in parallel with the historical experiment
but fixes aerosol and aerosol precursor emissions to pre-industrial conditions.
Therefore, the differences between these two experiments are attributable to
anthropogenic aerosol emissions. Note that we use the hist-piAer simulations but not



the hist-aer simulations designed by the Detection and Attribution Model
Intercomparison Project (DAMIP; Gillett et al., 2016), which resembles the historical
simulations but are only forced by transient changes in aerosol. The design of the
hist-piAer simulation means that it can also capture any nonlinearities resulting from
GHG-driven changes in clouds.
The monthly outputs from historical and hist-piAer simulations for ESMs are
used, including TAS, all-sky outgoing shortwave radiation at the top-of-atmosphere
(OSR), OSR assuming clear sky (OSRclr), mass mixing ratio of sulphate aerosol in
the atmosphere (mmrso4), total cloud amount (CLT), and cloud-top effective droplet
radius ($r_{eff}$). The corresponding lower-complexity models have conducted the
historical but not the hist-piAer simulations, and only the monthly TAS output from
historical simulations are used. Therefore, we focus on the ESMs, which allow a
simple way of diagnosing the sources of the anomalous cooling. The main variables are
summarized in Table 2.
The verification data used in this study is HadCRUT5, the monthly $5^{o}$lat by $5^{o}$lon
gridded surface temperature (Morice et al., 2021), a blend of the Met Office Hadley
Centre SST data set HadSST4 (Kennedy et al., 2019) and the land surface air
temperature CRUTEM5 (Osborn et al., 2021).
**2.3 Method**
By comparing the TAS anomalies in ESMs and the lower-complexity models
with HadCRUT5, our study found that TAS anomalies from 1960 to 1990 relative to
1850-1900 in ESMs and most of the lower-complexity models are on average much
lower than observed, resembling a "pot-hole" shape. This period of anomalous
cooling, i.e., the "pot-hole" cooling (PHC), is then quantified as the near-global mean
($60^{o}$S to $65^{o}$N) difference in the TAS anomaly between models and HadCRUT5 from
1960 to 1990. The variations over the polar regions (north of $65^{o}$N and south of $60^{o}$S)
are not considered due to the lack of long-term reliable observations (Wu et al., 2019a).
The PHC period coincides with a period when global emissions of $SO_2$, the main
precursor of sulphate aerosol, rapidly increased.



The aerosol cooling due to aerosol-radiation interaction (ARI) is dominated by
the contribution of sulphate aerosol as estimated by models and observations (-0.35$\pm$
0.5W m$^{-2}$ for the total ARI and -0.4$\pm$0.2W m$^{-2}$ for ARI of sulphate aerosol (Myhre et
al., 2013). We use the evolution of sulphate loading (loadSO4) through the historic
simulation as a proxy for total aerosol concentration changes to link estimates of the
impact of aerosol-forcing-sensitivity. Whilst the overall impact of aerosol forcing will
also be driven by other aerosol species, we adopt this approach because the sulphates
dominate estimates of aerosol-forcing-sensitivity during this period and other aerosols
species can be assumed (as a 1$^{st}$ order approximation) to have covaried with the SO$_2$
emissions during this period. As such when we present estimates of the aerosol
impact/loadSO4 we are presenting the impact of all aerosol species (including
absorbing aerosols such as black carbon) as they covary with the sulphate
concentrations during the historic period. The motivation for presenting it in this way,
is we can separate differences in ESM responses to changes in aerosol amount from
the differences in aerosol amount (represented by loadSO4) simulated by the ESMs.
We can estimate the impact of anthropogenic aerosol by using the difference in
OSR between the historical and hist-piAer simulations, *ΔOSR*. *ΔOSR* is of course
involves any differences in planetary albedo, between the two simulations, including
clear-sky albedo changes and any adjustments in microphysical or macroscopic
properties of clouds. The aerosol-forcing-sensitivity can be calculated from a linear fit
between the OSR differences and loadSO4 differences between the historical and
hist-piAer simulations (*ΔOSR /ΔloadSO4*). The linear fit slope measures the sensitivity
of total aerosol forcing. Wilcox et al. (2015) found a large diversity of the CMIP5
models in simulating the total aerosol forcing, which arises from the diversity in
global load and spatial distribution of sulphate aerosol, and differences in global mean
cloud top effective radius, amongst other factors. In this study, we diagnose the OSR
differences from historical simulations that also capture the temperature response.
As such the OSR differences do not represent a measure of only the aerosol forcing
impact but combine OSR differences arising from both the aerosol forcing and the





temperature response to this forcing, which we will refer to in this manuscript as the
aerosol-forcing-sensitivity. So, the numbers diagnosed here cannot be directly
compared to aerosol forcing estimates published elsewhere, but the relative magnitude
and time evolution of the aerosol-forcing-sensitivity is informative of the aerosol
radiative role in the simulations we explore here.
The aerosol-forcing-sensitivity can be further partitioned into a contribution from
aerosol-radiation interactions (ARI), and aerosol-cloud interactions (ACI). ARI
quantifies the influence of aerosols on clear-sky radiative fluxes. ACI is due to the
impact of aerosol-induced changes in the properties of clouds, such as cloud spatial
extent (amount), cloud longevity (lifetime), and cloud albedo on radiative fluxes. ARI
and ACI can be readily estimated from the CMIP6 output because annual mean cloud
amount, CLT, and the top-of-atmosphere radiative flux *assuming only clear-sky*,
OSRclr, are available for all the CMIP6 ESMs. For each model, the OSR from clouds
(OSRcld) can therefore be estimate as OSR-((1-CLT/100.)*OSRclr). As shown in the
Appendix, the aerosol-forcing-sensitivity can be expressed as:
$\overbrace{\textit{Aerosol-forcing-sensitivity}}$    $\overbrace{\textit{Aerosol-rad. Interactions (ARI)}}$    $\overbrace{\textit{cloud-amount term}}$
$\overbrace{\textit{ΔOSR/ΔloadSO4}}$   =   $\overbrace{\textit{(1-CLT\_hist/100)*M}}$    +    $\overbrace{\textit{(A-OSRclr\_hist/100.)*N}}$

*+ cloud-albedo term*     +    *residual*,         (1)

where CLT_hist and OSRclr_hist are the mean cloud amount (CLT) and clear-sky
OSR (OSRclr) in the historical simulation, and M, N and A are empirically
determined parameters. The parameter M is the slope of a linear fit of ΔOSRclr to
ΔloadSO4, and therefore measures the strength of the aerosol-radiation interactions in
each model. The term *(1-CLT_hist/100.)*M* can therefore be identified with ARI.
The parameter A is the slope of a linear fit of ΔOSRcld to ΔCLT, and therefore
measures the correlation of the short wave radiation reflected by clouds with changes
in cloud amount. The parameter N is the slope of a linear fit of ΔCLT to ΔloadSO4,
and therefore measures the sensitivity of cloud amount to aerosols. Note that changes
in cloud amount by definition also affect the fraction of clear-sky, hence increases in


OSRcld due to increases in CLT (i.e., *A\*N*) can be partly offset by changes in area of
clear-sky containing aerosols (OSRclr_hist\*N). The second term on the right-hand
side of Eq. (1) therefore contributes to the ACI, specifically it is the part of ACI that is
linearly proportional to changes to cloud fraction. It is roughly analogous to the "cloud
lifetime effect" (Albrecht, 1989), but is sensitive to *any* aerosol-induced cloud fraction
changes (Lohmann and Feichter, 2005), including any slow adjustments in clouds due
to feedbacks within the Earth System.

In addition to depending on ΔCLT, ACI is also influenced by any changes in

cloud-albedo that might occur independently of cloud-amount changes. Such
adjustments would include increases in simulated cloud-droplet effective radius
without accompanying changes in cloud cover. Changes purely in the brightness of
clouds, without changes in macroscopic properties of clouds, are difficult to identify
from the CMIP6 output because all the bulk-properties of clouds co-vary over the
course of the projections. However, subtracting ARI and the cloud-amount term from
the aerosol-forcing-sensitivity gives a residual that is, by definition, linearly
independent of cloud fraction differences (since by construction these have been
regressed out). This residual can then be interpreted as due to differences in the albedo
of clouds between the historical and hist-piAer, and will be called the "cloud-albedo
term". Note that this method of calculation implies that purely albedo effects cannot
be distinguished from general residual terms that result from the linear approximation
made.

Note that our decomposed ACI does not correspond exactly to the definitions of

"first" and "second" aerosol indirect effects. For example, the first indirect effect is
properly defined as variations of aerosol forcing when cloud droplet number
concentration varies at a constant value of the cloud liquid water path. This effect
cannot be isolated from the available CMIP6 output.

**3.   The "pot-hole" bias in CMIP6 ESMs**

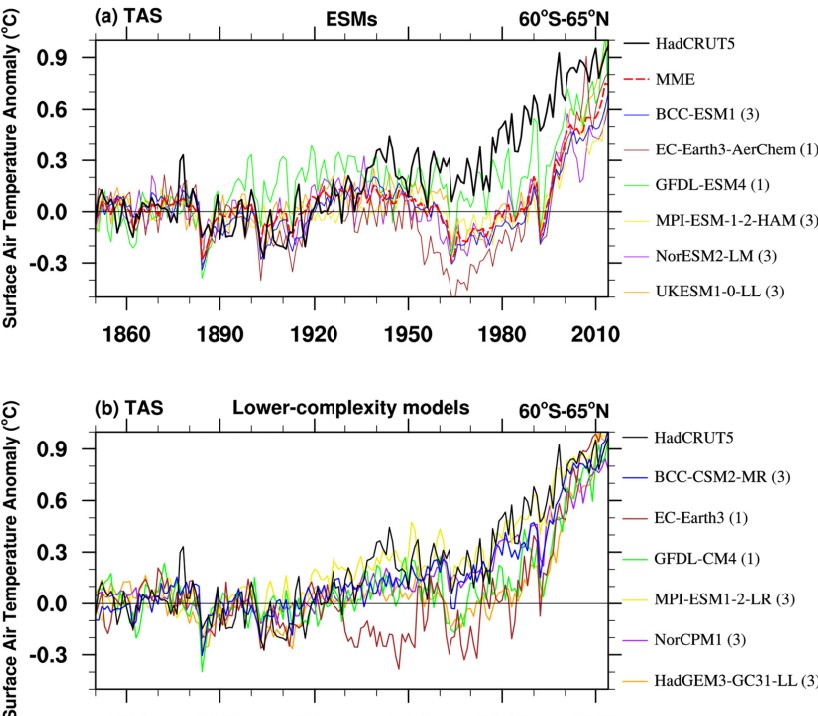

**Figure 1.** (a) Historical near-global mean (60°S to 65°N) surface air temperature (TAS) anomalies from HadCRUT5 (thick black line), the multi-member ensemble mean for each ESM (MMM, solid color lines), and their ensemble (MME, dashed red line). (b) is the same as (a), but for the lower-complexity models. The baseline is from 1850 to 1900. Units: °C. Value in bracket is the number of available members for each model.

Figure 1a shows the near-global averaged time series of annual mean TAS anomaly relative to 1850 to 1900 in HadCRUT5, the multi-member ensemble means (MMMs) for BCC-ESM1, MPI-ESM-1-2-HAM, NorESM2-LM and UKESM1-0-LL, and the first member for EC-Earth3-AerChem and GFDL-ESM4 during the historical period from 1850 to 2014. Only the first member for EC-Earth3-AerChem and GFDL-ESM4 is examined here, because only their first member is available for the hist-piAer experiement. The unforced, long-term drifts in TAS may occur in some of the ESMs, as estimated by their control simulation under pre-industrial conditions



(Yool et al., 2020). We have not accounted for long-term control simulation drifts in
our study as we are assuming that our focus on inter-decadal scale variability of TAS
anomalies is likely to be fairly insensitive to any century scale drifts.
The TAS anomaly in HadCRUT5 is generally above the baseline climate from the
1940s onwards, and warms fastest from the 1980s to 1990s. Compared with the
observations, all the ESM simulations have negative TAS anomaly biases after the
1940s, which are also evident in the ensemble-mean historical TAS of 25 CMIP6
models with and without interactive chemistry schemes (Flynn and Mauritsen, 2020).
In the ESMs and their ensemble mean (MME), the cold anomaly biases resemble a
"pot-hole" shape (Fig.1a), which is relatively small before the 1950s and after the
2000s but expands from the 1960s to 1990s. To reduce the impact of cooling responses
to the Pinatubo eruption in the early 1990s and the change in the spatial pattern of the
emissions, we mainly focus on the excessively cold anomaly from 1960 to 1990 in this
study. The period of anomalous cold in the global mean from 1960 to 1990 in model
simulations is defined as the "pot-hole" cooling (PHC) as described in section 2.3.
Table 3 shows the TAS anomaly biases in two typical periods, three decades before
the PHC period (1929~1959) and the PHC period (1960~1990). The biases are all
negative in the previous era; the negative biases increase in the PHC period and
become larger than -0.3 $^{\circ}$C in all the ESMs except for GFDL-ESM4. The PHC ranges
from -0.20$^{\circ}$C to -0.58$^{\circ}$C among the MMMs with a standard deviation of 0.11$^{\circ}$C, but
intra-model spread of PHC for the three available members in BCC-ESM1,
MPI-ESM-1-2-HAM, NorESM2-LM, and UKESM1-0-LL is relatively smaller. That
is, model structural uncertainty is more responsible for PHC than internal climate
variability.

**Table 3.** Biases in near-global averaged TAS anomalies relative to 1850-1900 from
the multi-member ensemble mean (MMM) and standard deviation across members
(SD) for each ESM and the corresponding lower-complexity model during
1929~1959, and the "pot-hole" period (1960~1990). The ensemble mean of MMMs



(MME) and the SD across MMMs are also examined. The anomalous cooling larger
than -0.3$^{\circ}$C is bolded. Biases are relative to the HadCRUT5.

| ESMs | 1929~1959 | 1960~1990 (PHC) | Lower-complexity models | 1929~1959 | 1960~1990 (PHC) |
|---|---|---|---|---|---|
| | MMM (SD) | MMM (SD) | | MMM (SD) | MMM (SD) |
| **BCC-ESM1** | -0.12 (0.01) | **-0.45** (0.07) | **BCC-CSM2-MR** | -0.09 (0.01) | -0.10 (0.01) |
| **EC-Earth-AerChem** | -0.27 | **-0.58** | **EC-Earth3** | **-0.37** | **-0.37** |
| **GFDL-ESM4** | -0.02 | -0.20 | **GFDL-CM4** | -0.12 | -0.26 |
| **MPI-ESM-1-2-HAM** | -0.16 (0.01) | **-0.39** (0.03) | **MPI-ESM1-2-LR** | 0.03 (0.03) | 0.01 (0.01) |
| **NorESM2-LM** | -0.16 (0.04) | **-0.41** (0.04) | **NorCPM1** | -0.10 (0.03) | -0.08 (0.04) |
| **UKESM1-0-LL** | -0.10 (0.09) | **-0.38** (0.08) | **HadGEM3-GC31-LL** | -0.16 (0.02) | **-0.33** **(0.03)** |
| **MME** | -0.14 (0.08) | **-0.40** (0.11) | | | |


The PHC bias is generally smaller in the corresponding lower-complexity models
(Fig.1b). BCC-CSM2-MR and MPI-ESM1-2-LR with prescribed chemistry and
aerosol can reasonably reproduce the TAS anomaly during the PHC period. The
anomalous TAS biases are about -0.10$^{\circ}$C in BCC-CSM2-MR and 0.01$^{\circ}$C in
MPI-ESM1-2-LR, which are also close to the biases in previous era (-0.09$^{\circ}$C in
BCC-CSM2-MR and 0.03$^{\circ}$C in MPI-ESM1-2-LR in 1929~1959). EC-Earth3 also
prescribes chemistry and aerosol but has a large PHC bias (-0.37$^{\circ}$C). The anomalous
cooling bias is also evident in previous era (1929~1959) with comparable amplitude.
However, the anomalous TAS biases in the second and third historical members of
EC-Earth3 during the PHC period are smaller and both positive (0.07$^{\circ}$C and 0.24$^{\circ}$C)
in our further examination. That is, TAS in EC-Earth3 may be sensitive to initial
condition and has been noted in Döscher et al. (2021). The PHC biases in GFDL-CM4





and HadGEM3-GC31-LL with prescribed chemistry and interactive aerosol scheme,
are comparable with that in the corresponding ESMs, but the biases grow slower from
previous era: -0.14 $^{o}$C in GFDL-CM4 v.s. -0.18 $^{o}$C in GFDL-ESM4, -0.17 $^{o}$C in
HadGEM3-GC31-LL v.s. -0.28 $^{o}$C in UKESM1-0-LL. The NorCPM1 also employs an
interactive aerosol scheme but has a small anomalous TAS bias (-0.08 $^{o}$C), which is
due to the overestimated tropical and southern hemispheric warming (Fig.2k).
Generally, the different behaviours seen in Fig.1 suggest that aerosol forcings may be
overestimated in the ESMs and the anomalous cooling in ESMs is a result of the extra
complexity associated with interactive chemistry and aerosol processes.
The evolution of zonal mean annually averaged TAS anomalies in HadCRUT5,
and the MMM for each ESM and lower-complexity model are further examined in
Fig.2. In HadCRUT5, TAS anomalies are generally positive after the 1940s. The most
significant TAS anomalies are evident in the late 20$^{th}$ Century and at the beginning of
the 21$^{st}$ Century, especially over the NH midlatitudes, where the TAS anomalies are
larger than 1.0 $^{o}$C. The results from BCC-CSM2-MR and MPI-ESM1-2-LR agree well
with the observations. However, the ESMs and the other lower-complexity models
simulate pronounced cold anomalies over NH subtropical-to-high latitudes during the
PHC period. Figure 2 also shows the evolution of surface anthropogenic $SO_2$
emissions (the contours). They rapidly increase during the PHC period. The latitudes of
the cooling centers are spatially co-located with the $SO_2$ emission sources – North
America and East Asia (at around 30$^{o}$N) and Western Europe (at around 50$^{o}$N). The
emission centers generally move south around the 1990s. European and North
American $SO_2$ emissions have reduced the since the 1980s; East Asian emissions
clearly increased from 2000 to 2005, followed by a decrease with large uncertainties
(Aas et al., 2020).



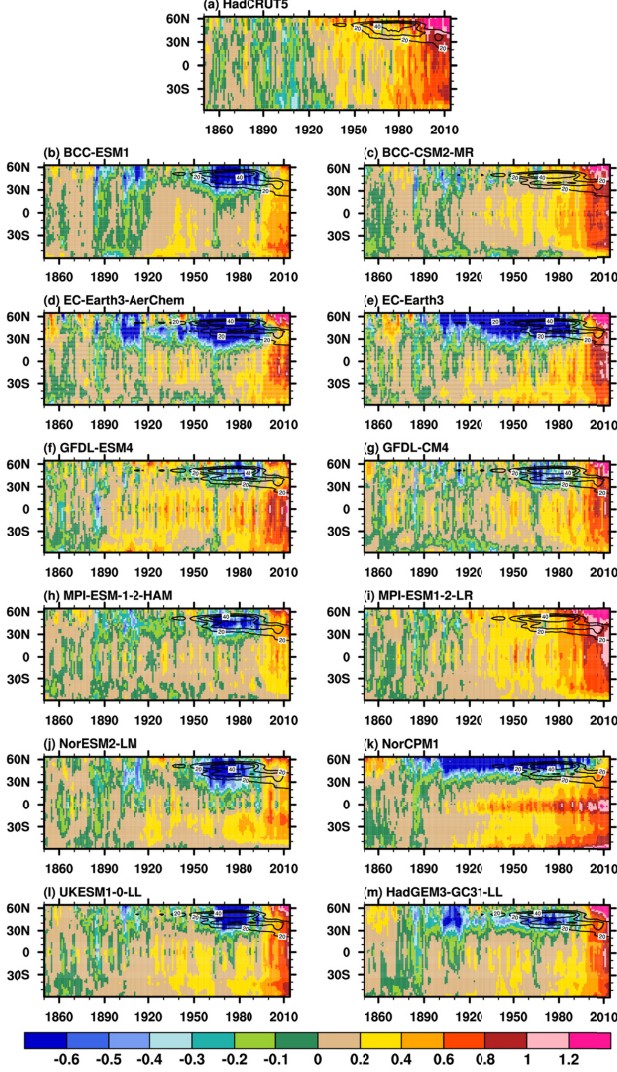

**Figure 2.** Time-latitude cross-section for annual-mean TAS anomalies (shaded) from (a) HadCRUT5, the MMM in each ESM (left panel), and the corresponding lower-complexity model (right panel). The anomalies are related to the 1850 ~ 1900 mean. Units: °C. Contours range from 20 to 40 ng m$^{-2}$ s$^{-1}$ with an interval of 10 ng m$^{-2}$ s$^{-1}$ show the zonal mean anthropogenic surface $SO_2$ emission provided by CMIP6.



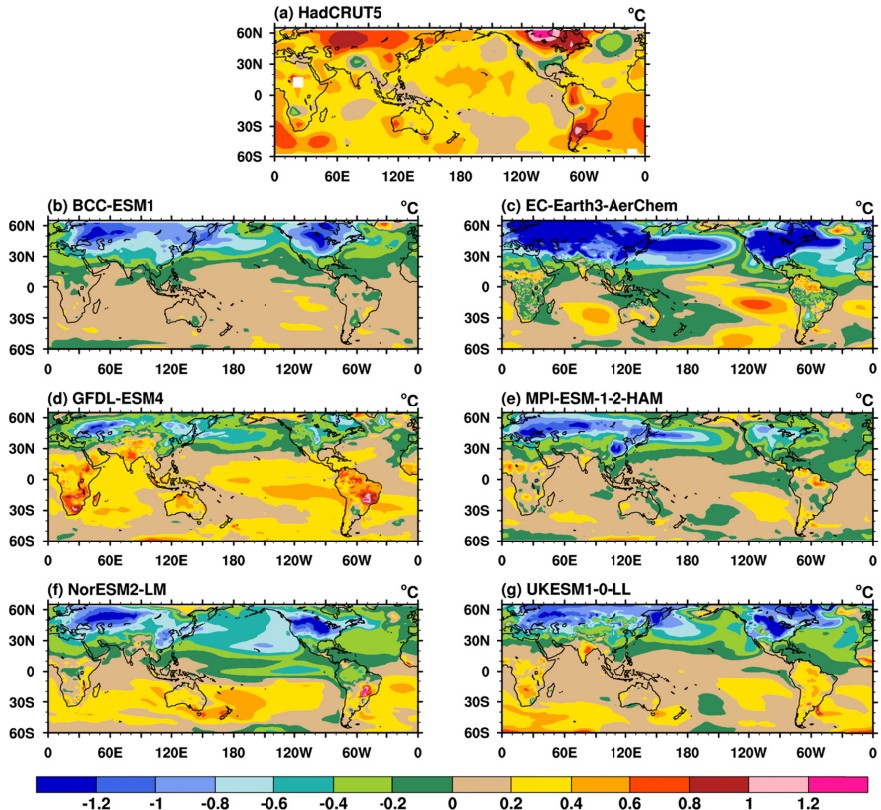

**Figure 3.** The TAS anomalies during the "pot-hole" period (1960 ~ 1990) from (a) HadCRUT5 and

(b-g) the MMMs in each of the ESMs. The anomalies are relative to the 1850~1900 mean. Units: °C.

Figure 3 examines the spatial structure of TAS anomalies in HadCRUT5 and

ESMs in the PHC period. The TAS anomalies in HadCRUT5 are generally positive and

are the largest over Eurasia and North America. The warm anomalies are on average

more than 0.4 °C along the 30°N ~ 60°N latitudinal belt. However, the ESMs show

anomalies with the opposite sign. The PHC is pronounced over major $SO_2$ emission

centers (Western Europe, East Asia, and the east US) and their downstream regions.

The cold anomalies over Eurasia and North America are lower than -0.6°C in the

ESMs.




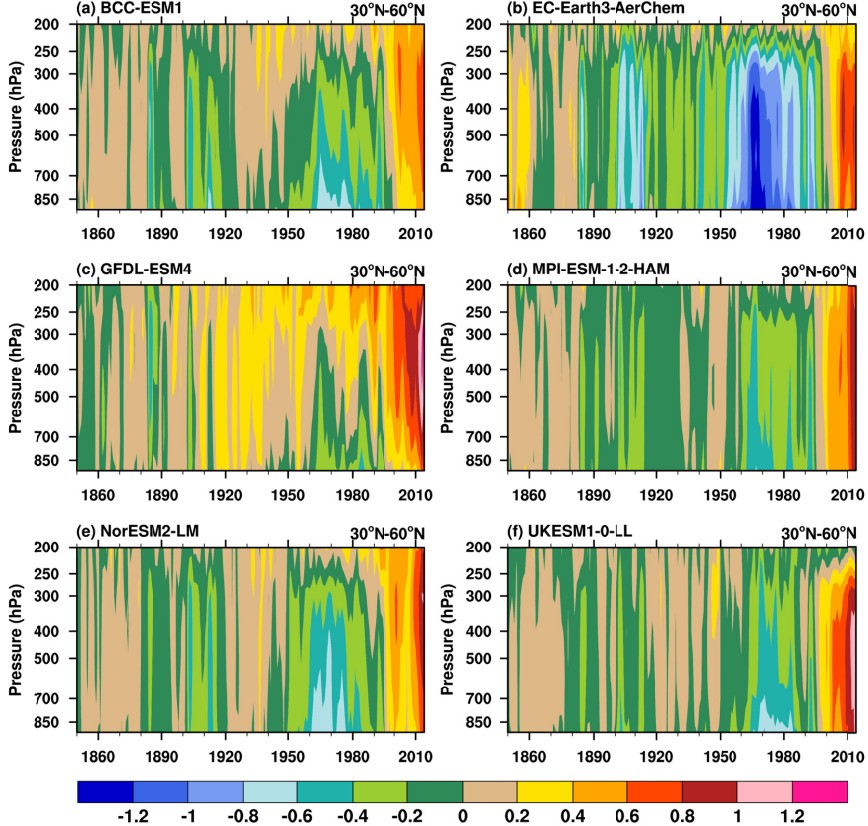


**Figure 4** Time-height cross-section of temperature anomalies averaged over the 30°N~60°N for the

MMM of each ESM. The anomalies are relative to the 1850 ~ 1900 mean. Units: °C.


The vertical structures of temperature anomalies over the 30°N~60°N are also

examined (Fig.4). The cold anomalies during the PHC period are the strongest at

lower levels and extend into the upper troposphere. This is distinct from the amplified

upper-tropospheric warming in the tropics in response to greenhouse gases (Thorne et

al., 2011). The cooling extends to higher altitudes in the troposphere when an

explosive volcanic eruption occurs, such as the 1963 Agung eruption, the 1974 Fuego

eruption, the 1982 El Chichon eruption, and the 1991 Mount Pinatubo eruption.

## 4. Possible reasons for the excessive cooling

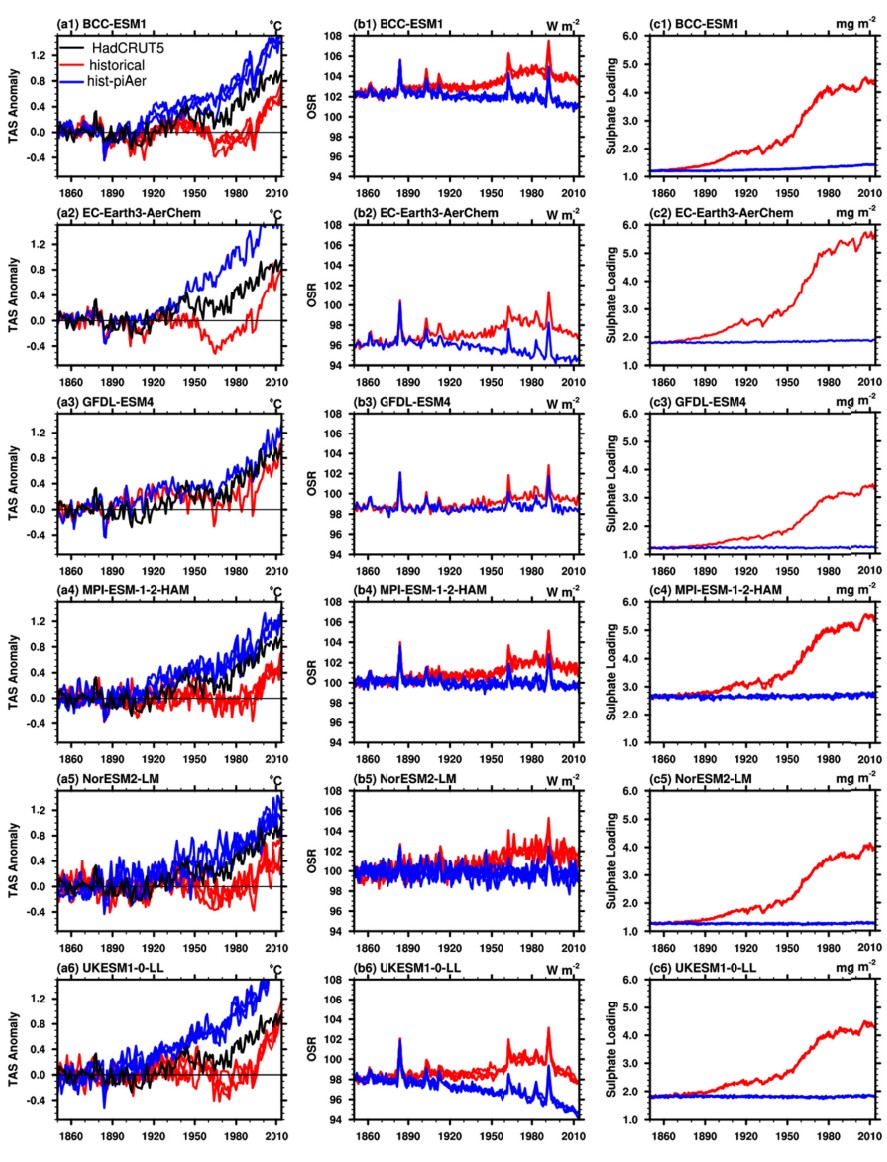


**Figure 5.** Evolutions of global annual means of (a1-a6) TAS anomalies (left panel, units: °C.), (b1-b6)
outgoing shortwave radiation at TOA (OSR, middle panel, units: W m$^{-2}$), and (c1-c6) sulphate loading
(right panel, units: mg m$^{-2}$) in HadCRUT5 (black line), each ESM member of the historical (red lines),
and hist-piAer experiments (blue lines). The TAS anomalies are relative to the 1850~1900 mean.




The differences between the historical and hist-piAer simulations help to
investigate the impact of anthropogenic aerosol emissions and its possible contribution
to the PHC biases. In this section, we examine the TAS, OSR, and sulphate loading
differences, and look in detail at their relationship. As shown by the evolution of TAS
anomalies in the two experiments (Fig.5, left panel), during the PHC period TAS
anomalies in HadCRUT5 (black line) are higher than those in the historical members
but lower than those in the hist-piAer members in all ESMs. That is, the model
responses to anthropogenic aerosol emission are larger than the amplitude of the PHC.
The temporal evolution of the OSR corresponds with that of the TAS but occurs in the
opposite direction (middle panel). The OSR differences between the historical and
hist-piAer simulations are larger in the ESMs that show big TAS differences (e.g.
EC-Earth3-AerChem and UKESM1-0-LL). The temporal evolution of the sulphate
loading (right panel) corresponds with the changes in anthropogenic surface $SO_2$
emissions (contours in Fig.2). Accordingly, the sulphate loading differences are
relatively small in the 19$^{th}$ Century, mildly increase in the first half of the 20$^{th}$
Century, grow most rapidly during the PHC period, and remain high afterward. In
comparison with the TAS and OSR differences, the intra-model spread of sulphate
loading for each ESM is relatively small. However, the inter-model diversity of
sulphate loading is large. For example, the sulphate loading difference between the
historical and hist-piAer experiments around the year 2000 is about 4 mg m$^{-2}$ in
EC-Earth3-AerChem, almost twice of that in GFDL-ESM4. With similar
anthropogenic $SO_2$ emission rates, the lower sulphate loading difference in
GFDL-ESM4 indicates it has a shorter sulphate aerosol residence time than that in
EC-Earth3-AerChem, which may be due to their different sulphate production and
deposition schemes. The large inter-model diversity is also evident in CMIP5 models
(Wilcox et al., 2015).
The latitudinal movement of the $SO_2$ emission center from the 1990s affects the
relative strength of aerosol forcing. Due to the more rapid oxidation at lower latitudes,



an equatorward shift in $SO_2$ emissions around 1990s result in a more efficient
production of sulphate and stronger aerosol forcing (Manktelow et al., 2007). The
northern mid-latitude temperature is also more sensitive to the distribution of aerosols,
which is approximately twice as large as the global average (Collins et al., 2013;
Shindell and Faluvegi, 2009). Therefore, we focus on the relationships between TAS,
OSR and sulphate loading after 1900 when $SO_2$ emissions changes are dominated by
its anthropogenic component, and before 1990 to reduce the effects of spatial changes
in anthropogenic $SO_2$ emission centers and the uncertainty of model response to the
1991 Mount Pinatubo eruption. As shown in Fig.6a, the TAS differences between the
historical and hist-piAer simulations vary linearly with the differences in the sulphate
loading for each ESM. The OSR differences are also linearly correlated with sulphate
loading differences for all models except UKESM1-0-LL (Fig.6b). It is interesting that
this nonlinearity is also observed in HadGEM2, a predecessor of UKESM1 (Wilcox et
al., 2015).



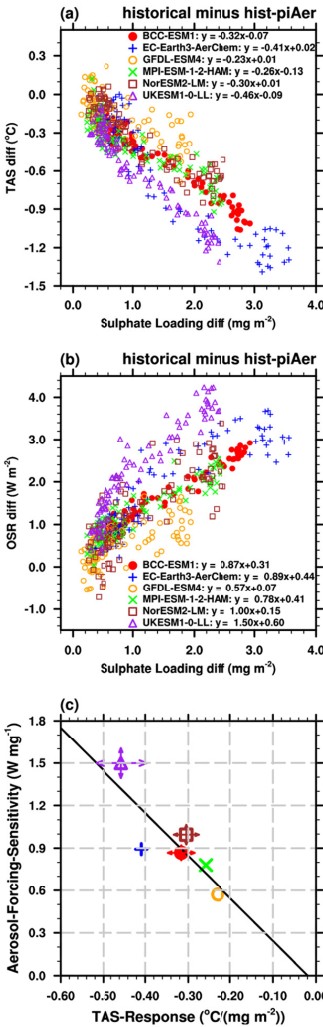

**Figure 6.** Scatters of 1900-1990 yearly sulphate loading differences between the historical and hist-piAer simulations (x-axis) versus (a) TAS differences and (b) OSR (y-axis). Results are from MMM in each ESM. The captions are the linear fitting equations. (c) shows the TAS response (x-axis) and aerosol-forcing-sensitivity (y-axis) which is equal to slope of linear fitting for each ESM (markers), and the corresponding intra-model spread (arrows).

The slope of the linear fitting equation between TAS (OSR) and sulphate loading as shown in the captions in Fig.6a (Fig.6b) is a measure of the sensitivity of TAS (aerosol forcing) to perturbations in atmospheric aerosol. Moreover, TAS-response





and aerosol-forcing-sensitivity are linearly correlated across the ESMs (Fig.6c). That
is, the strength of the TAS-response can be understood as the magnitude of
aerosol-forcing-sensitivity within each ESM. The similarities between the strength of
TAS-response and aerosol-forcing-sensitivity indicate the dominant role of the aerosol
cooling effect. The TAS-response and aerosol-forcing-sensitivity in UKESM1-0-LL
(the purple marker in Fig.6c) are the strongest, as well as their intra-model spread (the
length of arrows), indicating that TAS and aerosol forcing in this model are relatively
more susceptible to changes in aerosol. The TAS-response and
aerosol-forcing-sensitivity is the lowest in GFDL-ESM4.

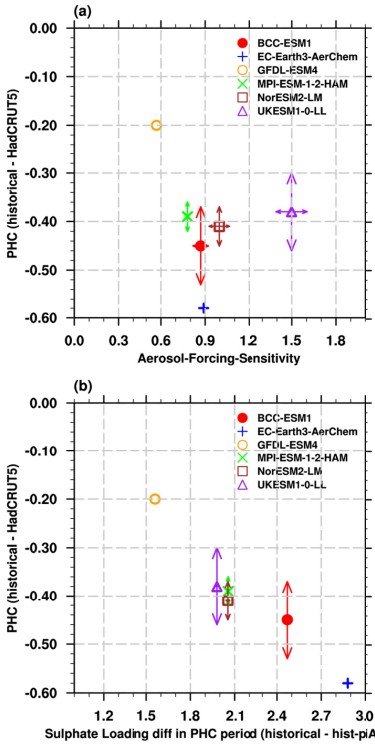

**Figure 7.** Pot-hole Cooling (PHC) bias in ESMs (°C) versus (a) the aerosol-forcing-sensitivity (W

428 mg$^{-1}$) and (b) sulphate loading differences (mg m$^{-2}$) during the PHC period. The arrows show the

429 uncertainty ranges among the members in each ESM.

428



428 The aerosol-forcing-sensitivity may be partly responsible for the PHC bias.

429 Figure 7a shows the PHC biases versus the aerosol-forcing-sensitivity (markers) and

430 their intra-model spread (arrows). The uncertainty of aerosol-forcing-sensitivity

431 (length of the horizontal arrows) in each ESM is smaller than the PHC bias uncertainty

432 (length of the vertical arrows). Despite the intra-model spread, the PHC and

433 aerosol-forcing-sensitivity seem to be negatively correlated. GFDL-ESM4 has the

434 weakest aerosol-forcing-sensitivity (~0.60 W mg$^{-1}$) and the smallest PHC (-0.20 $^{\circ}$C).

435 The amplitude of PHC in BCC-ESM1, MPI-ESM-1-2-HAM, and NorESM2-LM are

436 generally comparable, as is their aerosol-forcing-sensitivity. However, in

437 EC-Earth3-AerChem, the aerosol-forcing-sensitivity is close to those in BCC-ESM1,

438 MPI-ESM-1-2-HAM, and NorESM2-LM, but the PHC is more than 0.1$^{\circ}$C lower than

439 the others. In UKESM1-0-LL, the aerosol-forcing-sensitivity is the strongest (~1.5 W

440 mg$^{-1}$) but not the PHC bias. Therefore, the aerosol-forcing-sensitivity may be an

441 important contributor to PHC bias but cannot fully explain the inter-model diversity in

442 the PHC bias.

443 As shown by the X-axis in Fig.7b, the sulphate loading differences between the

444 historical and hist-piAer experiments during the PHC period are large among ESMs,

445 which are about 1.5 mg m$^{-2}$ in GFDL-ESM4 but approximately 2.9 mg m$^{-2}$ in

446 EC-Earth3-AerChem. Examination of the sulphate loading differences during the PHC

447 period and PHC biases shows that the PHC bias is generally larger in models with

448 higher sulphate loading over this period (Fig.7b). Therefore, the PHC biases may be

449 also attributable to sulphate loading related structural differences between ESMs.

451 **5. Discussion**

452 **5.1 The proportions of ARI and ACI**

453 There are significant differences in the aerosol-forcing-sensitivity among ESMs

454 (Fig.6b). The aerosol-forcing-sensitivity in UKESM1-0-L is almost three times of that

455 in GFDL-ESM4. Due to the uncertainties in physical processes and cloud

456 parameterizations, the dominant component (ARI or ACI) of



aerosol-forcing-sensitivity may also vary among ESMs. Here, we separate the different
components of the aerosol-forcing-sensitivity in each ESM by the method introduced
in the section 2.3 and Appendix. Sulphate loading is used as a proxy of aerosol amount
for all aerosol components in the quantification of the total effect because of its
dominant contribution to anthropogenic aerosol load during this period and its
covariation with the other aerosol species.

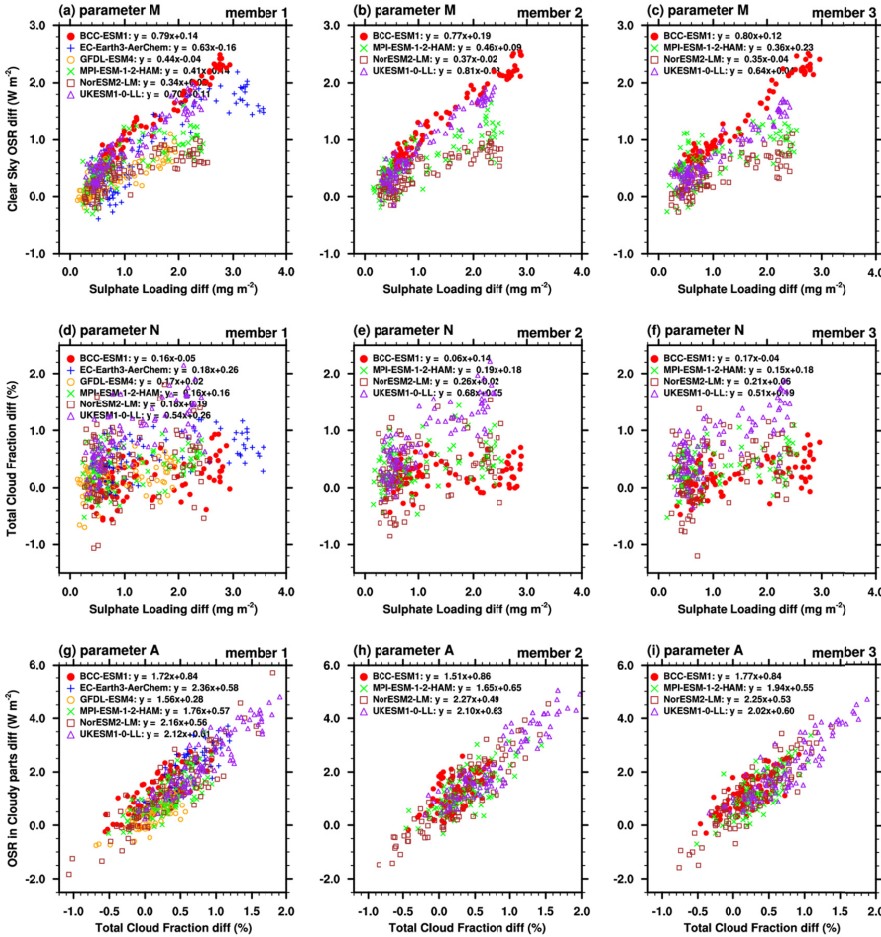

**Figure 8.** Annual mean differences between the historical and hist-piAer simulations in the ESM

members during 1900 to 1990 period for (a-c) sulphate loading (mg m$^{-2}$) versus clear-sky OSR (OSRclr,

W m$^{-2}$), (d-f) sulphate loading versus total cloud fraction (%), and (g-i) total cloud fraction versus OSR





470   in cloudy parts (W m$^{-2}$). Slopes of the linear fitting equations from the top row to the bottom row refer

to the parameters M, N, and A, respectively.

471

The ARI can be generally parameterized as (1-CLT_hist/100.)*M, where

CLT_hist is cloud amount in the historical simulation and parameter M is a measure of
the strength of aerosol-radiation interactions (*ΔOSRclr/ΔloadSO4*). Parameter M
varies widely from about 0.35W mg$^{-1}$ in NorESM2-LM to about 0.79 W mg$^{-1}$ in
BCC-ESM1 (captions in Fig.8a-8c). Since parameter M does not change much among
ensemble members in each ESM, their ARI is similar. That is, the impact of internal
climate variability on the ARI is relatively small, which is consistent with the
quantitative analysis in Fig.9 (Red bars).

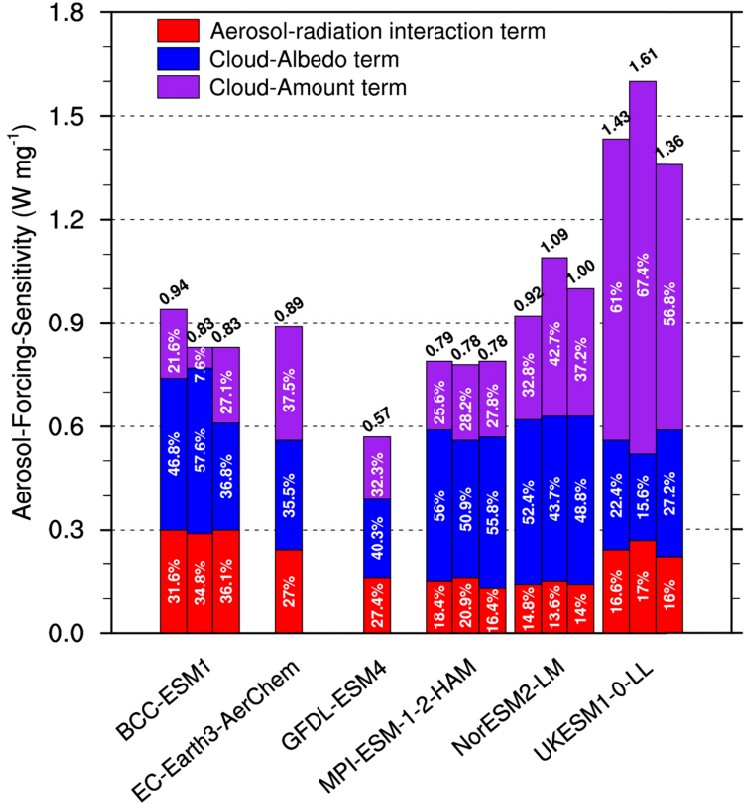



**Figure 9.** Total aerosol-forcing-sensitivity from each member in ESMs. The number marked on the top
is the total aerosol-forcing-sensitivity. Partition of aerosol-radiation interaction term, cloud-albedo term,
and cloud-amount term are marked in the corresponding color bars. Unit: W mg$^{-1}$.

The ACI can be estimated from the difference between the
aerosol-forcing-sensitivity and the ARI. The proportion of the
aerosol-forcing-sensitivity arising from the ACI is higher than 64% in all ESMs
(Fig.9). The inter-model variation of the ACI (0.37 W mg$^{-1}$) is much larger than that
for the ARI (0.09W mg$^{-1}$). For example, the ACI in UKESM1-0-LL (~1.2W mg$^{-1}$) is
higher than all the others and is about three times of that in GFDL-ESM4 (0.41 W
mg$^{-1}$). This demonstrates that differences in the aerosol-forcing-sensitivity across the
ESMs are dominated by the differences in their individual representation of ACI. The
intra-model variations in the ACI are also larger than that for the ARI. That is because
the intra-model variations of the ACI are influenced by the effects of climate system
internal variability on aerosol-induced cloud microphysics, with cloud radiative
properties and cloud lifetimes varying regionally. The intra-model variations are
attributable to the differences in atmospheric circulation among different ensemble
members, which may affect the geographical distributions of aerosols and clouds and
lead to a different magnitude of interactions.
The quantitative analysis in Fig.9 also indicates that ESMs with similar
aerosol-forcing-sensitivity may have different contributions from ARI and ACI. The
aerosol-forcing-sensitivity is similar in BCC-ESM1, EC-Earth3-AerChem,
MPI-ESM-1-2-HAM and NorESM2-LM, but the fractional contribution from the ACI
is the largest in NorESM2-LM and its ARI is less than half of that in BCC-ESM1.
Generally, BCC-ESM1 has the largest fractional ARI contribution (34%), whereas
NorESM2-LM has the largest fraction of ACI contribution (86%).

**5.2 The proportions of cloud-amount and cloud-albedo terms**



Our ACI metric includes several mechanisms by which aerosols can alter cloud

properties. This includes the cloud-albedo effects (or 'Twomey' effect), referred to as
the radiative forcing part of ACI, and effects of aerosols on the macroscopic properties
of clouds (for example, cloud extent and lifetime), referred to as the adjustments part of
ACI. However, it is complicated to separate these two parts of ACI directly using
available CMIP6 diagnostics, because the former is most accurately defined as a
change in cloud albedo with all other cloud properties held constant (i.e., a change in
cloud-droplet number concentration only), whilst the latter allows cloud properties to
respond.

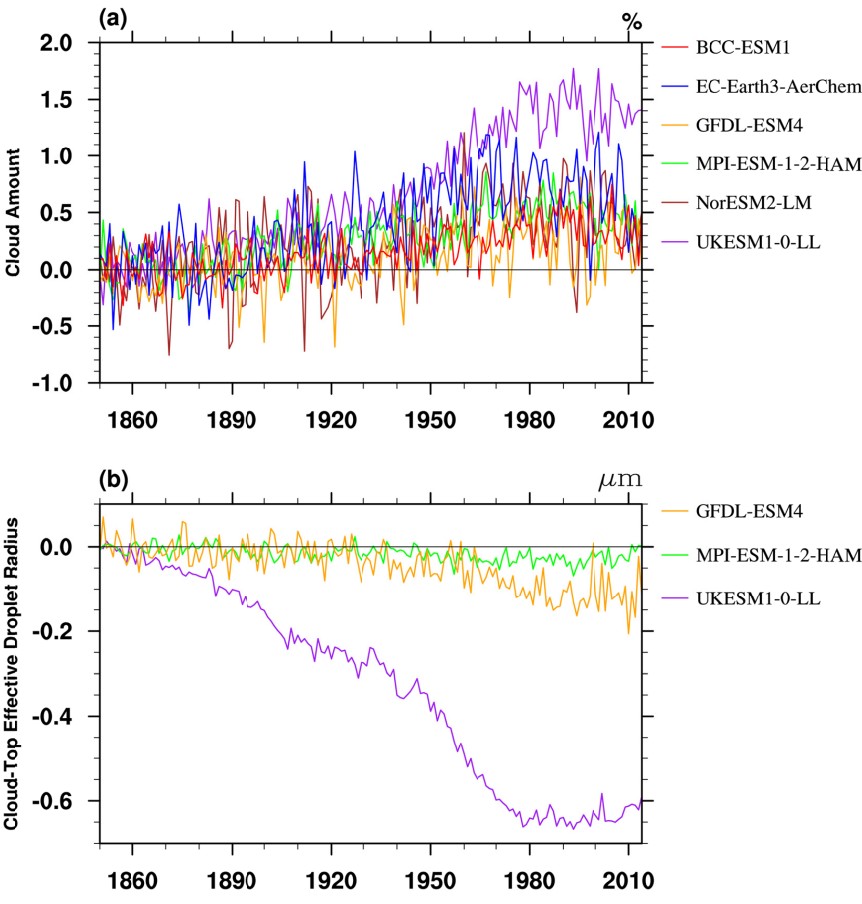



**Figure 10** (a) Evolutions of global mean cloud amount differences between the historical and
hist-piAer simulations in MMMs, units: %. (b) is the same as (a), but for cloud-top effective droplet
radius ($r_{eff}$, μm). The $r_{eff}$ data is only available for GFDL-ESM4, MPI-ESM-1-2-HAM, and
UKESM1-0-LL.

Figure 10 shows the evolution of global-mean differences in total cloud amount

($\Delta$CLT) and cloud-top effective droplet radius ($\Delta r_{eff}$) between the historical and
hist-piAer experiments. The $\Delta$CLT and $\Delta r_{eff}$ in UKESM1-0-LL are the largest and
highly correlated with each other (with a correlation coefficient of -0.93). The $\Delta r_{eff}$ and
$\Delta$CLT are generally related to the radiative forcing part and adjustments part of ACI,
respectively (Albrecht, 1989; Twomey, 1991). Therefore, the radiative forcing part and
adjustments part of ACI may be closely coupled in UKESM1-0-LL and are hard to
separate statistically. The strong correlation between cloud amount and $r_{eff}$ response in
UKESM1-0-LL indicates that this model is sensitivity to aerosol-cloud interactions,
which is likely to contribute to it having the strongest aerosol-forcing-sensitivity of all
the CMIP6 models in Fig.6b.

However, it is still possible to split the ACI into a part that is correlated with cloud

amount differences and a residual term. This can be done statistically by regressing-out
the approximate linear dependence of the differences between historical and hist-piAer
simulations of cloudy part OSR (OSRcld_p) on cloud fraction in each ESM (parameter
A in Fig.8g-8i). We call the degree of linear correlation of $\Delta$OSRcld_p with $\Delta$CLT the
"cloud-amount term", and the residual will be referred to as the "cloud-albedo term".
However, we reiterate that the so-called "cloud-amount term" may also include
changes in the reflectivity of clouds if these are correlated with changes in cloud
amount. Similarly, the cloud-albedo term will contain any sources of cloud amount
changes which have not been removed by linearly regressing OSRcld_p against cloud
amount. As such, we do not intend this nomenclature to indicate a precise separation of
the radiative forcing part and adjustments part of ACI. Our decomposition allows first
order assessment of these terms from historical simulations without the need for extra



simulations or calls, and also allows estimates from observations and intermodel
comparisons.
As described in the section 2.3 and the Appendix, the cloud-amount term is
sensitive to two parameters: the cloud amount response (parameter N in Fig.8d-8f)
and the sensitivity of OSR reflected from clouds to cloud amount changes (parameter
A, Fig.8g-8i). As shown in Fig.9, UKESM1-0-LL has the largest contribution of the
cloud-amount term to aerosol-forcing-sensitivity (62%, 0.91W mg$^{-1}$); the
cloud-amount term is the smallest in GFDL-ESM4 (~0.18W mg$^{-1}$). The cloud-albedo
term is defined to be linearly independent of cloud-amount changes (adjustments). For
the CMIP6 ESMs, it can only be estimated as the residual after subtracting the
cloud-amount term from the ACI. The cloud-albedo term is similar in BCC-ESM1,
MPI-ESM-1-2-HAM, and NorESM2-LM. The inter-model variation for the
cloud-amount term is about twice of that for the cloud-albedo term (0.29W/mg v.s.
0.16W/mg). That is, the variations of cloud-amount term are the major source of
inter-model ACI (and the aerosol-forcing-sensitivity) differences between ESMs.
Therefore, difference in the cloud-amount terms, across the ESMs, dominates the
uncertainties in the aerosol-forcing-sensitivity.
Note that, neither do our definitions correspond to the effects measured by using
multiple calls to the radiation scheme of a model, with and without aerosols, which
measure instaneous radiative effects; multiple calls give a measure of the fast response
of clouds to aerosol perturbations in a fixed thermodynamic and dynamical background,
allowing for a clear separation between ACI and rapid adjustments (e.g., Bellouin et
al., 2013). This differs from aerosol forcing diagnosed by differencing climate
projections with different aerosol forcings, which include the slow effects of other
feedbacks. For example, differences in climate forcings can lead to different SST
patterns, which in turn alter the location and characteristics of clouds. Despite these
differences, an advantage of our classification is that it provides a possible method for
model evaluation since the variables used are also, in principle, available from the
observations.




### 6. Conclusion

581 This study focuses on the reproduction of historical surface air temperature anomalies in six CMIP6 ESMs. The ESMs systematically underestimate TAS anomalies relative to 1850 to 1900 in the NH midlatitudes, especially from 1960 to 1990, the "pot-hole" cooling (PHC) period. In the global mean, the excessive cooling in models is more pronounced at the surface, which is distinct from the response to greenhouse gases that preferentially heat the tropical upper troposphere. Previous studies suggested that aerosol cooling is too strong in many CMIP6 models. Our study more specifically found that the PHC is concurrent in time and space with anthropogenic $SO_2$ emissions, which rapidly increase in the PHC period in NH. The primary role of aerosol emissions in these biases is further supported by the differences between ESMs and the lower-complexity models. Differences between historical simulations and simulations with aerosol emissions fixed at their preindustrial levels (hist-piAer) are used to isolate the impacts of industrial aerosol emission. We propose that the overestimated aerosol concentrations and aerosol-forcing-sensitivity in the ESMs account for the spurious drop in TAS in the mid-twentieth century.

597 A large inter-model spread in the aerosol-forcing-sensitivity is evident between CMIP6 models. A simple metric is derived for determining the dominant contribution to the aerosol-forcing-sensitivity in any specific model. The ARI has a slight intra-model variation. The ACI accounts for more than 64% of the aerosol-forcing-sensitivity in all analyzed ESMs. The considerable inter-model variation in the aerosol-forcing-sensitivity is mainly attributable to the uncertainty in the ACI within models. The ACI can be further decomposed into a cloud-amount term and a cloud-albedo term. The cloud-amount term is found to be the major source of inter-model diversity of ACI.

606 Although the TAS anomaly is systematically underestimated in all ESMs, the reasons for the PHC are different among models. Therefore, different models require





different improvement strategies. For example, modifying sulphate deposition
processes may reduce the cold biases in EC-Earth3-AerChem, which has a relatively
large sulphate loadings; BCC-ESM1 has a relatively large proportion of ARI and may
need to focus on the effect of aerosol backscatter; the cloud-amount term (adjustments
part of ACI) contributes to more than 60% of the aerosol-forcing-sensitivity in
UKESM1-0-LL, which is comparable or even larger than the total
aerosol-forcing-sensitivity in the other ESMs. Considering the crucial role of cloud
properties on the inter-model spread in aerosol-forcing-sensitivity, the aerosol-cloud
interactions should be a focus in development of aerosol schemes within ESMs.

In this study, we mainly focus on the ESMs since all of them have the hist-piAer

experiment, which allows a simple way of diagnosing the sources of the anomalous
cooling and estimates the aerosol-forcing-sensitivity. The method to estimate the
aerosol-forcing-sensitivity can also be applied to the lower-complexity models.
Therefore, if the hist-piAer experiments were available for lower-complexity models,
it would be possible to isolate the contributions to aerosol-forcing-sensitivity that is
due to the added aerosol complexity in the ESMs.




**Appendix: Decomposition of the Aerosol-radiation interaction and aerosol-cloud interaction**

Considering the dominate role of sulphate aerosol on anthropogenic aerosol forcing, we use the sulphate loading (loadSO4) as a proxy for all aerosol in our analysis. The aerosol-forcing-sensitivity (as determined by the difference between the historical and hist-piAer experiments) is estimated by the all-sky OSR differences per sulfate burden unit ($\Delta$OSR/$\Delta$loadSO4) and it is the combination of OSR differences in the clear-sky parts ($\Delta$OSRclr_p/$\Delta$loadSO4) and the cloudy parts ($\Delta$OSRcld_p/$\Delta$loadSO4):

$\Delta OSR/\Delta$loadSO4 $= \Delta OSRclr\_p/\Delta$loadSO4 $+ \Delta OSRcld\_p/\Delta$loadSO4. (A1)

The OSRclr_p for a particular experiment can be calculated as:

$OSRclr\_p = (1-CLT/100.)*OSRclr,$ (A2)

where CLT is the total cloud amount (unit: %), and OSRclr is the OSR assuming all clear sky (unit: W/m$^2$). The cloud amount changes ($\Delta CLT$) will modify the propotion of clear-sky and then affect the OSR changes attributed to the clear-sky part by covering or uncovering aerosols in clear sky. Therefore, based on equation (A2), $\Delta$OSRclr_p/$\Delta$loadSO4 can be decomposed into the OSRclr-response ($\Delta OSRclr/\Delta loadSO4$) and CLT-response ($\Delta CLT/\Delta loadSO4$):

$\Delta OSRclr\_p/\Delta loadSO4 = (1-CLT\_hist/100.)*\Delta OSRclr/\Delta loadSO4$

$- OSRclr\_hist/100*\Delta CLT/\Delta loadSO4+residual\_clrp$

$= (1-CLT\_hist/100.)*M - OSRclr\_hist/100*N + residual\_clrp,$ (A3)

where CLT_hist and OSRclr_hist are the mean CLT and OSRclr in the historical experiment. Residual_clrp is the residual term that is non-linear in $\Delta$OSRclr and $\Delta$CLT. The parameter M= $\Delta OSRclr/\Delta loadSO4$ is related to strength of aerosol-radiation interaction and can be estimated by linear fitting of $\Delta$OSRclr on $\Delta$loadSO4. The parameter N= $\Delta CLT/\Delta loadSO4$ is related to CLT-response and estimated by linear fitting of $\Delta$CLT on $\Delta$loadSO4.

The OSRcld_p is the cloudy part of OSR, accounting for the difference between OSR and OSRclr_p. The cloudy part of the OSR differences ($\Delta$OSRcld_p) can be generally estimated as:





*ΔOSRcld_p = A\*ΔCLT + cloud-albedo relative changes +residual_cld,*

where the parameter A $=Δ(OSR-OSRclr\_p)/ΔCLT$ is the sentivity of the shortwave
flux reflected by clouds to changes in cloud amount. The parameter A depends on the
baseline cloud albedo (radiative flux per cloud amount unit) and can be estimated by
linear fitting of $ΔOSRcld\_p$ on $ΔCLT$. Hence,

*ΔOSRcld_p/ΔloadSO4 = A\*ΔCLT/ΔloadSO4 + cloud-albedo term*

*+residual_cld,*

*= A\*N + cloud-albedo term +residual_cld,*    (A4)


where N is the parameter defined above. The cloud-albedo term on the right-hand side
of equation (A4) can be obtained as a residual after subtracting *A\*N* from *ΔOSRcld_p/*
*ΔloadSO4*, thereby eliminating any linear dependence of the cloudy-sky shortwave
flux response on cloud-amount changes.
As with the clear-sky decomposition, *residual_cld* is a possible non-linear term
and is assumed to be small. This term cannot in fact be distinguished from the
cloud-albedo term, in this analysis: we must therefore accept that cloud-albedo
changes could be accompanied by non-linear changes in macroscopic cloud properties
(in this framework).

The total aerosol-forcing-sensitivity can be measured by substituting the
derived values of $ΔOSR/ΔloadSO4$ from both the clear sky (equation A3) and
cloudy (equation A4) parts back into equation (A1):

*ΔOSR/ΔloadSO4*     *= (1-CLT_hist/100.)\*M - OSRclr_hist/100\*N*

*+A\*N + cloud_albedo_term + residual*

*= (1-CLT_hist/100)\*M + (A - OSRclr_hist/100.)\*N*

*+ cloud_albedo_term + residual_osr.*    (A5)


Based on equation (A5), the total aerosol-forcing-sensitivity can therefore be
decomposed to the aerosol-radiation interaction term (ARI), (1-CLT_hist/100.)\*M,
cloud-amount term as (A - OSRclr_hist/100.)\*N, and cloud-albedo term (defined as a
residual).


**Data Availability.** All the model data can be freely downloaded from the Earth System
Federation Grid (ESGF) nodes (https://esgf-node.llnl.gov/search/cmip6/). The global
historical surface temperature anomalies HadCRUT5 dataset is freely available on
https://www.metoffice.gov.uk/hadobs/hadcrut5/data/current/download.html.

**Author contributions**

The main ideas were formulated by JZ, KF, STT, JPM, and TW. JZ, KF, and STT
wrote the original draft, and the results were supervised by LJW, BBB, and DS. All
the authors discussed the results and contributed to the final manuscript.

**Competing interests**

The authors declare that they have no conflict of interest.

**Acknowledgments**

This work was supported by The National Key Research and Development Program
of China (Grant no. 2018YFE0196000 and 2016YFA0602100). All the AUTHORS
were supported by the UK-China Research & Innovation Partnership Fund through the
Met Office Climate Science for Service Partnership (CSSP) China as part of the
Newton Fund. LJW was supported by the National Environmental Research Council
(NERC) "North Atlantic Climate System Integrated Study" (ACSIS) program.



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
