# Peer review of "The role of anthropogenic aerosols in the anomalous cooling from 1960 to 1990 in the CMIP6 Earth System Models"

_Atmospheric Chemistry and Physics, 2021_

## Author Comment (AC1)

In the following, the text with italicization indicates the Reviewers' comments, and the normal text is our response.

**Replies to Reviewer's comments:**

Thank you very much for the insightful comments, which lead to a significant improvement. Here is the item-by-item reply to your comments.

*Reviewer(s)' Comments to Author(s):*

*Reviewer: 1*

*General Comments:*

*This manuscript, "The role of anthropogenic aerosols in the anomalous cooling from 1960 to 1990 in the CMIP6 Earth System Models," investigates the causes of the mid-century excessive surface air temperature (TAS) cooling in the CMIP6 earth system model (ESM) ensemble relative to observations, what the authors have dubbed the "pot-hole" cooling (PHC). Internal variability does not explain the anomalous cooling. This study links the PHC bias to anthropogenic SO2 emissions (as a proxy for all aerosols, because sulphates are the dominant aerosol in this time period), which are much larger in the ESMs during the PHC period than in observations. The PHC is also most pronounced over the Northern Hemisphere midlatitude sources of sulphates during this period, further supporting the connection between the anomalous cooling in models and exaggerated aerosol emissions over North America, East Asia, and Europe within ESMs. The PHC is further attributed to differences in the sensitivities of the ESMs to changes in aerosol loading, modulated through the impact of aerosol changes on outgoing shortwave radiation at top-of-atmosphere (OSR), called the aerosol-forcing-sensitivity; change here refers to the difference between the historical simulation for each ESM, and the hist-piAer simulation, which is identical to the historical simulation except that aerosol emissions are held fixed at preindustrial levels. Impacts of aerosols on cloud amount in particular were found to be the major driver of inter-model spread in aerosol-forcing-sensitivity, and thus the PHC effect.*

1. *This manuscript would benefit from more focusing of the main results and somewhat less attention to all the details, except where necessary to describe and support the main conclusions. It is somewhat easy to get lost in the descriptions of results and lose sight of the main takeaways, and some sections would benefit from being worded more concisely, such as the paragraph beginning at Line 282. The section describing Figures 2-4 could also be shortened; Fig. 4 doesn't*

*seem to add any new information that is critical to the conclusions, and so could be dropped from the manuscript. Fig. 2 likewise may not be necessary to include or could be replaced by an additional subplot in Fig. 1 showing the time series of SO2 emissions or sulphate loading (for the ESMs at least); Fig. 3 seems enough to link the PHC spatially to the centers of aerosol emissions and contours of anthropogenic SO2 emissions could be added here as they were for Fig. 2. And it should be made clearer that the lower-complexity models in these plots support the results for the ESMs concerning exaggerated sulphate loading relative to the observations.*

*Response:* You are right. Following modifications have been made:

*(1)* The section describing the main features of PHC biases (section 3) is shortened and worded more precisely.

*(2)* We delete the plots of vertical TAS anomalies (Fig.4), and only keep the main description (L377-L379): "The PHC biases are strongest at lower levels (Figures not shown), distinct from the response to greenhouse gases."

*(3)* Fig.1 shows the time series of global mean TAS anomalies to identify the anomalous cooling biases in ESM. However, the cooling biases may be offset or amplified by the biases over other regions as in NorCPM1. The main purpose of Fig.2 is to further examine the anomalous TAS and present the close relationship between the PHC biases and the SO2 emission both temporally and spatially. Due to the zonal advection, the cooling biases during the PHC period are evident along the latitudes of emission centers. So, the close relationship between the PHC biases and the SO2 emission is seen more clearly on the time-latitude plot. There are nine figures in the updated manuscript, so we think it may be alright to keep Fig.2.

*(4)* As suggested, we clarify that the lower-complexity models support the results for the ESMs (L350-353): "Generally, the different behaviours seen in Fig.1 and Fig.2 suggest that aerosol forcings may be overestimated in the ESMs and lower-complexity models with interactive aerosol scheme, and the anomalous cooling is a result of the extra complexity associated with aerosol processes." L373-374: "However, the ESMs show anomalies with the opposite sign (Fig.3b-3g), as do the lower-complexity models with interactive aerosol scheme (figures not shown)."

2. *The larger issue in this paper is with the formulation of the aerosol-forcing-sensitivity and its decomposition in aerosol-radiation interactions (ARI) and aerosol-cloud interactions (ACI). Lines 160-162 and lines 594-596, for example, either state or imply that the impact of differences in aerosol amount within the ESM (overestimated aerosol loading) and the impact of ESM response to aerosol amount changes (aerosol-forcing-sensitivity) have been separated from each other and their impact on temperature response quantified. However, the manuscript does not clearly do so, nor clearly justifies the decomposition. The*

*aerosol-forcing-sensivity defined as ΔOSR /ΔloadSO4 is clearly useful, as shown, for example, by its high correlation with the change in temperature per unit change in sulphate loading in Fig. 6c, but this does not seem to easily translate into a high correlation with the PHC (Fig. 7a), and which is the main focus of this analysis. Indeed, Fig. 7b seems to show the opposite of the main conclusions of this manuscript: the PHC difference between historical and hist-piAer experiments is much more strongly correlated with the change sulphate between the historical and hist-piAer simulations, while the aerosol-forcing-sensitivity in Fig. 7a does not seem to show the negative correlation claimed in Line 433; the aerosol-forcing-sensitivity does not seem to explain the inter-model spread in PHC bias. The correlations presented in Figure 8 between OSR_clearsky and total cloud fraction with sulphate loading, in combination with Figures 2-4, point clearly to the impacts of overestimation of the aerosol loading in the models, but does not really separate it into a forcing-sensitivity. The unclear separation between these two aerosol impacts (concentrations and forcing impacts) need to be further developed and justified before the conclusions can be considered more firm, or the text and figures better clarified if already sufficiently developed.*

**Response:**

You are right. By comparing the historical and hist-piAer experiments, we found the approximately linear response of TAS to aerosol loading (Fig.5a) and the impact of aerosol-forcing-sensitivity on the TAS response in ESMs (Fig.5c). In the decomposition, we try to quantify the relative contribution of aerosol loading and aerosol-forcing-sensitivity to the PHC biases. As shown in Fig.6a, the PHC biases in MPI-ESM, NorESM2, and UKESM1 are about -0.40°C, but the aerosol-forcing-sensitivity ranges from 0.78 to 1.5 W mg$^{-1}$. The aerosol-forcing-sensitivity in UKESM1-0-LL is the strongest (~1.5 W mg-1) but not the PHC bias. On the other hand, a negative correlation is evident between the aerosol loading and PHC during the PHC period (Fig. 6b). That is, there is a strong relationship between the PHC biases and the overestimated aerosol concentration, but the aerosol-forcing-sensitivity cannot well explain the inter-model PHC diversity. In the modified manuscript we emphasize that the differences in aerosol loadings amongst the ESMs contribute to the PHC biases. The effect of aerosol-forcing-sensitivity may also account for some of the model differences, such as the small aerosol-forcing-sensitivity in GFDL-ESM4, but the contribution is smaller (L463-484, description and discussion about Fig.6).

In our examination on the aerosol-cloud-interaction (ACI), the results in MPI and UKESM are discussed and we suggest that the different way models treat ACI may be a reason to some of the discrepancies in the aerosol-forcing-sensitivity related to PHC in Fig.6a  (L575-583): "The strong correlation between cloud amount and reff response in UKESM1-0-LL indicates that this model is sensitive to aerosol-cloud interactions, which is likely to contributes to it having the strongest aerosol-forcing-sensitivity and intra-model spread of all the CMIP6 models in Fig.5c. MPI-ESM-1-2-HAM and UKESM1-0-LL have similar ensemble mean

PHC biases and close sulphate burden, but the aerosol-forcing-sensitivity differences in UKESM1-0-LL is almost twice of that in MPI-ESM-1-2-HAM (Fig.5). That is, the overestimated sulphate burden dominates the PHC biases, but the ACI sensitivity may partly affect the amplitude and uncertainty ranges of PHC biases."

3. ***This leads to the formulation of the aerosol-forcing-sensitivity into ARI and ACI in Equation 1 and in the appendix. The variables used in this formula, OSR, SO4 loading, and cloud amount, do not seem to be independent of each other (as in Fig 8), but are treated as independent variables. This raises some doubts about the validity of the linear decomposition presented here, and further makes it seems as if Equation 1 is an over-regression of the overestimated aerosol concentrations onto the radiative fluxes in the models.***

**Response:** The formulation of Equation 4 (manuscript) and Equation A5 (appendix) is intended to separate the two potential factors that can lead to larger temperature responses to aerosol emissions. The first term is a representation of the model differences in the number of aerosols simulated in the atmospheric column despite the use of a common emission inventory. The second term represents the model differences in the response of clouds and their impact on radiation to changes in aerosol amount. Equation 4 is intended to illustrate how an estimate of the latter can be separated from the former. If model differences in aerosol loading were primarily responsible for the simulated temperature biases then strong correlations would exist between these variables and appear in the first term. However, if the model differences in the temperature response were being driven by the impact of aerosols on the radiation response from clouds then this would appear in correlations within the aerosol-forcing-sensitivity term. We find that model differences in the PHC period temperature response is primarily driven by differences in the simulated aerosol loading.

There may be potential interactions that we do not explicitly account for in this formulation. For example, it has been suggested that models with higher aerosol loading will tend to produce a weaker aerosol-cloud responses because the

aerosols would be providing a greater buffering leading to a less sensitive response (e.g. Carslaw et al., 2013, https://doi.org/10.1038/nature12674). Whilst this non-linearity is likely to be important at the regional scale, at a global mean scale the there is evidence to suggest that the forcing may be a more linear response to global emission changes   (e.g. Booth et al, 2018, https://doi.org/10.1175/JCLI-D-17-0369.1, Kretzschmar et al, 2017 https://doi.org/10.1175/JCLI-D-16-0668.1). So, our formulation (equation 1 in the revised manuscript) explicitly assumes that there is a broadly linear relationship between loadSO4 and emissions, and OSR with loadSO4 (and the linearity or otherwise of cloud amount would influence the latter) and any interaction is small. This is an important assumption, which we clarify in L255-265:

"We acknowledge that the linear approximation in our method doesn't explicitly account for the absorption above clouds, or the adjustments to aerosol-radiation interactions (e.g., Carslaw et al., 2013) that are known to be locally important. Our formulation explicitly assumes that there is a broadly linear relationship between loadSO4 and emissions, and aerosol radiation with loadSO4 (and non-linearity due to cloud albedo or amount or any interaction is small at global scale as suggested in Booth et al. (2018)). Should these interaction terms be non-negligible in this analysis, we still expect the broader attribution of the reasons for the model diversity in temperature response over the PHC period, either how they simulate aerosol concentrations or how they simulate the response to this, to generally hold."

4. *The manuscript would also benefit from a much clearer explanation of the origin of the terms and their combination in the appendix (and therefore Equation 1): which term corresponds to which process exactly, and so why they are included in the various steps of the derivation of Equation 1 presented in the appendix. This would also help explain how Equation 1 is different from simply being the effect of overestimated aerosol loading, and how ARI and ACI are differentiated from each other. Again, this need to be further developed or more clearly explained and justified to firm up the conclusions of this manuscript.*

**Response:** Equation 4 (manuscript) and Equation A5 (appendix) are intended to explore differences when the leading order impact of model differences in aerosol amount, is removed. The "aerosol-forcing-sensitivity" is the radiative impact after normalizing these model differences by differences in their atmospheric loadings. Any differences in this "aerosol-forcing-sensitivity" term emphasis differences in the direct radiative effects (as would be found in clear sky conditions) or the aerosol effects on clouds. This helps isolate where model differences might be coming from (i.e. are they due to aerosol-radiation interaction, changes in cloud amount, or changes in cloud properties). However, we acknowledge that this is an approximation designed to be used with existing simulations, rather than a strict decomposition.

We explain each term in Eq.A3, Eq.A4, and Eq.A5 in the appendix as suggested and include a scaled down version in the main text to take the reader through its derivation (L200-239). We also improve the description of aerosol-forcing-sensitivity in the method section (L173-176): "The sensitivity of OSR-response to aerosol changes, i.e., the aerosol-forcing-sensitivity, can be measured by the linear fit slope between the annual mean globally averaged OSR differences and loadSO4 differences between the historical and hist-piAer simulations."; compare it with the definition of aerosol effective radiative forcing (L188-192): "The aerosol-forcing-sensitivity is different from the commonly used aerosol effective radiative forcing (ERFaer), which is the change in net TOA downward radiative flux after allowing adjustments in the atmosphere, but with sea surface temperatures and sea ice cover are fixed at climatological values."; and compare the ACI and ARI term in L249-255: "Decomposition of the ARI, the cloud-amount term and cloud-albedo term of ACI are detailed further in the Appendix. The aerosol-cloud feedbacks are mainly in the ACI term which includes cloud spatial extent (amount), cloud albedo on radiative fluxes, and cloud particle swelling by humidification (Christensen et al., 2017; Neubauer et al., 2017). There is also a (smaller) effect of feedback on the ARI term that is also affected by cloud amount

changes insofar as increased/decreased cloud cover can obscure/reveal clear-sky radiative fluxes."

*Specific Comments:*

1. *Lines 260-263: Volcanic forcing has been left in, is that right? How are other major eruptions, like Agung, within the 1960s-1990s period treated?*

   **Response:** Yes, the impacts of the other eruptions are considered in our study because it is hard to eliminate the impacts of volcanic forcing within the PHC period only by the historical simulations. We focus on the period before 1990 mainly considering the spatial changes of emissions. We rewrite the sentences to make it clear (L303-309): "To reduce the impact of the change in the spatial pattern of the emissions in the late 20th century, and the Pinatubo eruption in the early 1990s, we mainly focus on the excessively cold anomaly from 1960 to 1990 in this study. The impacts from the Agung (1963) and El Chichon (1982) eruptions have been left in the PHC period as their effect on the simulated temperature is not as pronounced as the response to Pinatubo and are short-lived in time compared to the period we study."

2. *Lines 165-167: Do differences between the two simulations in planetary albedo, clear-sky albedo, etc. need to be accounted for when decomposing aerosol-forcing-sensitivity? Do they complicate interpretation of the results presented in this paper, and why/why not?*

   **Response:** Differences in the planetary and clear sky albedo will influence the radiative response to a given change in aerosol loading. This will be one of several factors that will determine whether a particular model produces a stronger or weaker aerosol-forcing-sensitivity. In the analysis we are using the difference between the historical simulation and the hist-piAer simulation so that the radiative differences presented in this paper reflect those arising only in response to the

aerosol changes. Differences in planetary and clear sky albedo will be just a few of the factors that would influence the magnitude of these diagnosed terms.

In this study, we do not take the further step to further break down the causes of the different aerosol-forcing-sensitivity, beyond the decomposition into the ARI and ACI components

3. *Lines 170-173: Wilcox et al. (2015) seems out of place, and needs to be more clearly related to the methods/results discussed here.*

   **Response:** Agree. The reference is deleted.

4. *Lines 394-398: This is really a repeat from Lines 260-263, so this sentence here is unnecessary.*

   **Response:** Agree. The redundant part is deleted.

5. *Line 401: Any indication why for UKESM, or are the reasons still unknown?*

   **Response:** We think the strong aerosol-cloud albedo effect may account and add the statement in L429-435: "In both cases, the approximation of linearity holds less well for UKESM1-0-LL, especially at small sulphate loadings. This reflects the behaviour of HadGEM2, a predecessor of UKESM1 (Wilcox et al., 2015), and is likely to be due to the strong aerosol-cloud albedo effect in these models. The global mean annual mean reff decreases by about 0.7 μm since pre-industrial era, more than twice the magnitude of change seen in the other models (Fig.1b in Wilcox et al., 2015 and Fig.9b in this study)."

6. *Lines 526-563: What about the correlations for the other two models that provided effective droplet radius output? It would also be very interesting to contrast UKESM to MPI, since they have similar PHC biases but different aerosol-forcing-sensitivities – how much does the sensitivity matter if they produce the same temperature bias with different sensitivities, or is it due to differences in ARI or ACI (or just differences in aerosol loading)?*

[Figure]

*Figure A1.* *Annual mean differences between the historical and hist-piAer simulations in the ESM members during 1900 to 1990 period for cloud-top effective droplet radius (reff, μm) versus total cloud fraction (%). The numbers follow model names are the correlation coefficients.*

[Figure]

*Figure A2.* *Scatter plot of 1900-1900 yearly sulphate loading differences between the historical and hist-piAer simulations in MPI-ESM-1-2-HAM (x-axis) versus in UKESM1-0-LL (y-axis).*

*Response:*

The cloud-top effective droplet radius (reff) differences are highly correlated with the total cloud fraction differences in UKESM with correlation coefficient of 0.92. As shown in Fig.A1, the correlations are -0.40 for MPI and insignificant for GFDL (-0.09). The differences between historical and hist-piAer experiments in MPI and GFDL models are also much smaller than in UKESM, especially for the Δreff. We clarify this in L568-571: "For the other two ESMs for which Δreff was archived, the correlation coefficient is -0.40 for MPI-ESM-1-2-HAM and insignificant for GFDL-ESM4 (-0.09). The ΔCLT and Δreff differences are smaller in MPI-ESM-1-2-HAM and GFDL-ESM4 than in UKESM1-0-LL, especially for the Δreff differences."

You are right. Comparisons between UKESM and MPI are very interesting since they have similar PHC biases but different aerosol-forcing-sensitivities. As shown in Fig.A2, the sulphate loading differences in UKESM and MPI are very close, and the ratio is nearly one. We demonstrate the dominant role of aerosol loading on PHC biases (as for the response to General Comment 2) and the impact of ACI by comparing the results in UKESM and MPI (L575-583): "The strong correlation between cloud amount and reff response in UKESM1-0-LL indicates that this model is sensitive to aerosol-cloud interactions, which is likely to contributes to it having the strongest aerosol-forcing-sensitivity and intra-model spread of all the CMIP6 models in Fig.5c. MPI-ESM-1-2-HAM and UKESM1-0-LL have similar ensemble mean PHC biases and close sulphate burden, but the aerosol-forcing-sensitivity differences in UKESM1-0-LL is almost twice of that in MPI-ESM-1-2-HAM (Fig.5). That is, the overestimated sulphate burden dominates the PHC biases, but the ACI sensitivity may partly affect the amplitude and uncertainty ranges of PHC biases."

*Technical Corrections:*

1. *Line 151: Need second closing parenthesis.*

   **Response:** Done. Sentence has been modified.

2. *Line 151: Need to add the variable loadSO4 to Table 1.*

**Response:** In the ESGF node, not all the ESMs upload the loadSO4 for historical and hist-piAer experiments. Therefore, in our study, loadSO4 is calculated from the mass mixing ratio of sulphate aerosol (mmrso4). So loadSO4 is described and listed as a variable in Table 1 but not a CMIP6 diagnostic label.

3. *Line 164: Don't need the "is"*

   **Response:** Deleted.

4. *Line 190: Should say "be estimated"*

   **Response:** Done.

5. *Line 628: Should say "dominant"*

   **Response:** Done.

6. *Fig. 6: Caption should say "Scatter plots"*

   **Response:** Done.

---

## Author Comment (AC2)

In the following, the text with italicization indicates the Reviewers' comments, and the normal text is our response.

**Replies to Reviewer's comments:**

Thank you very much for the insightful comments, which lead to a significant improvement. Here is the item-by-item reply to your comments.

*Reviewer(s)' Comments to Author(s):*

*Reviewer 2:*

*Zhang et al. investigate the role of aerosol forcing in CMIP6 models for a period in which temperatures were too low compared to observations. The study is important and diligently performed. It is of large interest to the readers of Atmos Chem Phys and mostly well written.*

*I have only two major comments, but a number of specific ones.*

*Major remarks*

1. *It would be necessary to list the aerosol effective radiative forcing for each model, as defined from the fixed-SST simulations. This could be in Table 1 or 3; it would be useful to discuss this in comparison to the transient diagnostics.*
*Response:* **Done.** We list the ERFaer for each ESM in Table 3 and discuss their relationship with the transient diagnostics (the aerosol-forcing-sensitivity) in our study in L188-L192: "The aerosol-forcing-sensitivity is different from the commonly used aerosol effective radiative forcing (ERFaer), which is the change in net TOA downward radiative flux after allowing adjustments in the atmosphere, but with sea surface temperatures and sea ice cover are fixed at climatological values." and in L454-456: "The aerosol-forcing-sensitivity is not correlated with the aerosol effective radiative forcing (ERFaer, Table 3), largely due to the strong influence of UKESM1-0-LL on the result."

2. *The term A in Eq. 1 is wrong (A = 1 in general). This has consequences for the analysis in Section 5.*
*Response:* Parameter A estimates the sensitivity of the shortwave flux reflected by clouds to changes in cloud amount. Therefore, the parameter A depends on the baseline cloud albedo (radiative flux per cloud amount unit) and is calculated as $\Delta$(OSR-OSRclr_p)/$\Delta$CLT in our study: $\Delta$ means differences between historical and hist-piAer simulations; OSR-OSRclr_p is the cloudy part OSR; CLT is the total cloud amount.

Therefore, as estimated in the bottom plots in Fig. 7, parameter A varies among ESMs and generally ranges from 1.5 to 2.4. We include a scaled down version of the decomposition in the main text to take the reader through the derivation (L174-238) and try to explain parameter A more clearly in L221 to L223: "That is, the parameter A generally represent the baseline cloud albedo which is sensitive to the cloud parameterizations relative to Cloud Droplet Number Concentration (CDNC), cloud-droplet effective radius, and the other factors."

***Specific remarks***

**l22 The term "aerosol-forcing-sensitivity" is not a standard term and I don't understand it at this point. Either the abstract needs to define it or manage without this new term.**

*Response:* **Done.** We added the definition of "aerosol-forcing-sensitivity" in the abstract (L19-21): "The aerosol-forcing sensitivity, estimated as the outgoing shortwave radiation (OSR) response to aerosol concentration changes, cannot well explain the diversity of PHC biases in the ESMs." We also try to better clarify it in section 2.3 (L173-186): "The sensitivity of the OSR-response to aerosol changes, i.e., the aerosol-forcing-sensitivity, can be measured by the linear fit slope between the annual mean globally averaged OSR differences and loadSO4 differences between the historical and hist-piAer simulations"

***l45 It might be wise to make clear that greenhouse gases accumulate, aerosols do not at time scales longer than a week.***

*Response:* **Done**. We add the statement in L43-44: "Aerosols are generally not evenly distributed around the planet as greenhouse gases, and they have relatively short lifetimes of the order of a week."

***l48 The authors need to clarify where the end point of this sentence is. In the most recent years, anthropogenic aerosol emissions clearly decreased.***

*Response:* **Done.** We add the statements about emission in recent years from L46 to L54: "The rate of change of global aerosol emissions slowed down in the late 20th century (Hoesly et al., 2018), and the trend of global emission has been negative since the mid-2000s (Klimont et al., 2013). There has also been a shift in emission source regions. European and US emissions have declined following the introduction of clean air legislation since the 1980s, while Asian emissions have risen due to economic development. East Asian emissions clearly increased from 2000 to 2005, followed by a decrease with large uncertainties (Aas et al., 2020; Wang et al., 2018). The decade long emission reduction since 2006 over East China is not well represented by the CMIP6 emission (Wang et al., 2021)."

***l50 But Chinese ones reached a peak already in 2011 and declined sharply since then.***
*Response:* You are right. Since the historical run in CMIP6 end in 2014, we cite Wang

et al. (2021) about the decade long emission reduction in China in L51-L54: "East Asian emissions clearly increased from 2000 to 2005, followed by a decrease with large uncertainties (Aas et al., 2020; Wang et al., 2018). The decade long emission reduction since 2006 over East China is not well represented by the CMIP6 emission (Wang et al., 2021)."

*l89 should read "van Noije"*
*Response:* **Done.** Modified in L93 and Table 1.

*l90 Mauritsen et al. (2019) is not the appropriate reference for MPI-ESM-1-2-HAM*
*Response:* **Done.** The reference for MPI-ESM-1-2-HAM is changed to Neubauer et al. (2019).

*l153 This term "aerosol-forcing-sensitivity" is introduced here the first time and needs to be defined precisely.*
*Response:* **Done.** We modify "aerosol-forcing-sensitivity" to "aerosol forcing" in this sentence which is more appropriate (L157 and L160). The definition of "aerosol-forcing-sensitivity" is given in the next paragraph (L173-186): "The sensitivity of the OSR-response to aerosol changes, i.e., the aerosol-forcing-sensitivity, can be measured by the linear fit slope between the annual mean globally averaged OSR differences and loadSO4 differences between the historical and hist-piAer simulations."

*l156 The authors need to provide a reference for this statement, or demonstrate by other means that it applies. It is not obvious, since the organic aerosol emissions often occur in very different places and are only partly linked to the fossil fuel burning that generates most anthropogenic SO2.*
*Response:* **Done.** You are right. The estimates of organic aerosol emissions are less available. In CMIP6 simulations, the emissions in Community Emissions Data System (CEDS) inventory are adopted by CMIP6 models. The CEDS emissions show similar evolutions for SO2, BC, and OC (Hoesly et al, 2018), especially for the steadily increasing in the mid-20$^{th}$ century (the pole-hole cooling period). We cite the reference about CEDS as suggested in L157-L163: "Whilst the overall impact of aerosol forcing will also depend on other aerosol species, we adopt this approach because the sulphates dominate estimates of aerosol forcing during this period and other aerosols species can be assumed (as a 1$^{st}$ order approximation) to have covaried with the SO$_2$ emissions during this period as presented by the Community Emissions Data System (CEDS) inventory adopted by CMIP6 models (Hoesly et al, 2018)."

*l164 drop "is"*
*Response:* **Done.**

*l167 And of course natural variability*
*Response:* **Done.** Modified in Line170-173: "$\Delta OSR$ of course involves any differences in natural variability and planetary albedo between the two simulations, including clearsky albedo changes and any adjustments in the microphysical or macroscopic properties of clouds."

*l169 It would be useful to clarify at which scale the data are aggregated. Are these monthly or annual means? global or regional means? If little aggregation, can the change in sulfate load not become very small?*

*Response:* We use annual mean globally averaged OSR and loadSO4 differences to estimate the aerosol-forcing-sensitivity. We clarified this in L173-L176: "The sensitivity of the OSR-response to aerosol changes, i.e., the aerosol-forcing-sensitivity, can be measured by the linear fit slope between the annual mean globally averaged OSR differences and loadSO4 differences between the historical and hist-piAer simulations."

*l170 Forcing or effective forcing?*
*Response:* Sentence is deleted.

*l177 It would be very useful to disentangle the two. The aerosol effective forcing is readily defined by the fixed-SST simulations designed for this purpose. The authors could investigate this in comparison to the same period in the runs they investigate here. Another option would be to make use of the DAMIP simulation with varying aerosol.*

*Response:* **Done (as the response to Major Remark 1).** We compare the "aerosol-forcing-sensitivity" in our study and the "aerosol effective forcing (ERFaer)" calculated from the fixed-SST data provided in RFMIP and shown in Table 3 of the manuscript.

It would be very interesting to compare with the DAMIP simulation with varying aerosol (hist-aer). However, there is only a small overlap between models participating in AerChemMIP (for the hist-piAer simulation) and those participating in DAMIP. Of the ESMs used in this study for the analysis of hist-piAer, only GFDL-ESM4 has done the hist-aer simulation.

*l183 To which extent is it possible to include the feedbacks in this distinction? Are the feedbacks mainly in the ACI term in this definition?*
*Response:* The aerosol-cloud feedbacks are mainly in the ACI term which includes cloud spatial extent (amount), cloud albedo on radiative fluxes, and cloud particle swelling by humidification. There is also a (smaller) effect of feedback on the ARI term it is also affected by cloud amount changes. We add the statement in L250-255: "The aerosol-cloud feedbacks are mainly in the ACI term which includes cloud spatial extent (amount), cloud albedo on radiative fluxes, and cloud particle swelling by humidification (Christensen et al., 2017; Neubauer et al., 2017). There is also a (smaller) effect of feedback on the ARI term that is also affected by cloud amount changes insofar as increased/decreased cloud cover can obscure/reveal clear-sky radiative fluxes."

The small and large effects of feedback in the ARI term and the ACI term are evident in the amplitude of ARI differences and the ACI (Cloud-Albedo term PLUS Cloud-Amount term) differences between ESMs (Fig. 8). This is consistent with Chen et al. (2014). We cite Chen et al. (2014) in L530-L532: "Chen et al. (2014) also

suggested that ACI is the main contribution to the Aerosol radiative forcing uncertainty and the response of marine clouds to aerosol changes is paramount."

***l184 This is an approximation that makes several mistakes. It neglects the (regionally very important) absorption above clouds, and it also neglects the adjustments to aerosol-radiation interactions.***
*Response:* Agreed. These effects might not be proportional to cloud fraction or clear-sky OSR, so they would appear as residuals in this framework. However, there is a suggestion in the literature that aerosol forcing and emissions may respond more linearly at a global mean scale than they are known to do so at regional scales (e.g. Booth et al, 2018, https://doi.org/10.1175/JCLI-D-17-0369.1, Kretzschmar et al, 2017). Our formulation explicitly assumes that there is a broadly linear relationship between loadSO4 and emissions, OSR and loadSO4, and non-linearity due cloud albedo or amount or any interaction is small. We now acknowledge this is in the text on L255-265: "We acknowledge that the linear approximation in our method doesn't explicitly account for the absorption above clouds, or the adjustments to aerosol-radiation interactions (e.g., Carslaw et al., 2013) that are known to be locally important. Our formulation explicitly assumes that there is a broadly linear relationship between loadSO4 and emissions, and aerosol radiation with loadSO4 (and non-linearity due to cloud albedo or amount or any interaction is small at global scale as suggested in Booth et al. (2018)). Should these interaction terms be non-negligible in this analysis, we still expect the broader attribution of the reasons for the model diversity in temperature response over the PHC period, either how they simulate aerosol concentrations or how they simulate the response to this, to generally hold."

***l186 Lifetime is only measurable in terms of horizontal and/or cloud albedo.***
*Response:* Good point! In this framework, cloud lifetime effects will show up via the albedo and areal extent changes. For simplicity, we've removed mention of 'cloud lifetime' from this list (L236).

***l200 This seems to be wrong. Instead of A, one should use OSRcld_hist in Eq. 1.***
**Response:** We think this is right (please see our derivation in Appendix A for details). The decomposition is a bi-linear regression, where we first regress out the linear dependence on clear-sky OSR difference (which gives the first term, and Parameter M, in Eq. 4) and then regress out linear dependence on cloud-amount differences (which gives the Parameter A). Parameter A is therefore the estimated radiative-flux change per cloud amount unit. We use the cloudy part OSR and cloud amount differences between historical and hist-piAer simulations( $\Delta$ (OSR-OSRclr_p)/$\Delta$CLT) to calculate parameter A. As estimated by the slope of linear fit in Fig.7g-7i, parameter A varies among ESMs and generally ranges from 1.5 to 2.4.

***l208 This is not true, firstly because a lifetime effect may also involve changes to cloud albedo, and second because this lifetime effect is (by far) not the only influence***

*on cloud extent (as the authors immediately acknowledge).*

**Response:** Agreed – we've removed the claim that the ACI-interaction term is 'analogous' to the Albrecht effect.

*l214 Presumably, "increases in cloud droplet number concentration"? In fact, this is the key impact of aerosols on clouds (Twomey effect, radiative forcing due to aerosol-cloud interactions). But adjustments of cloud water path are also included.*

*Response:* You are right. We add the impact of cloud droplet number concentration in L236 to L239: "Such adjustments would include increases in cloud droplet number concentration and increases in simulated cloud-droplet effective radius without accompanying changes in cloud cover."

*l223 Maybe it is noteworthy that, e.g., Chen et al. (Nature Geosci 2014) or Christensen et al. (ACP 2017)*

*Response:* These two papers are very interesting and support our study. They have been cited in L250-L253: "The aerosol-cloud feedbacks are mainly in the ACI term which includes cloud spatial extent (amount), cloud albedo on radiative fluxes, and cloud particle swelling by humidification (Christensen et al., 2017; Neubauer et al., 2017).", and L530-532: "Chen et al. (2014) also suggested that ACI is the main contribution to the Aerosol radiative forcing uncertainty and the response of marine clouds to aerosol changes is paramount."

*l226 In fact, the decomposition is so far off "first" and "second indirect effects" (and, by the way, the terms are obsolete since AR5) that it is better to drop this paragraph.*

*Response:* Agree. The paragraph is deleted.

*l241 Anomalies with respect to which time period average?*

*Response:* The anomalies are relative to 1850-1900 mean. We have rewritten the figure capture for Fig.1 to make it clear.

*l242 The terms "MMM" and "MME" seem strange to me. Often, MMM is multi-model mean, but here it seems it is, in contrary, the single-model ensemble mean. What is a "multi-member ensemble"? is that not tautological? Is the term "MMM" necessary at all? Why not simply "for each model, the ensemble mean is shown"? What does the acronym "MME" stand for?*

*Response:* Thank you for your explanation. I modified the statements about single-model ensemble mean and now "MMM" stand for the multi-model mean.

*l255 What distinguishes the "first" member from any member?*

*Response:* The member used here is actually the first realization for each model. We rewrite the sentence in L287-291: "Figure 1a shows the near-global averaged time series of annual mean TAS anomaly relative to 1850 to 1900 in HadCRUT5 during the historical period from 1850 to 2014, and the ensemble means for each model except for EC-Earth3-AerChem and GFDL-ESM4 (where only a single realization is available for

the hist-piAer experiment).”

*l292 "Sensitive to initial condition" seems strange wording for an influence of internal variability.*
*Response:* **Done.** We modified the wording as "The PHC bias are large (-0.37ºC) in EC-Earth3, which has prescribed chemistry and aerosol. The large bias may be a reflection of the large internal variability on TAS in EC-Earth3 (Döscher et al., 2021), for which we have only one member." in L329-L332.

*l320 A colour scale that evenly is distributed in positive and negative directions should be chosen. Or else this discrepancy should be pointed to in the caption.*
*Response:* **Done.** Since the warming amplitude in early 2000s is about twice of the amplitude of "pot-hole" cooling, we choose to use unevenly distributed color scale. We declare differences in the color scale in the caption: "Note that the color scale intervals in the positive and negative directions are 0.2 ºC and -0.1 ºC, respectively."

*l370 This is only true at a very superficial glance. There is not intimate link evident.*
*Response:* **Done.** We delete this sentence.

*l374 This correspondence is hardly evident.*
*Response:* **Done.** The northern-hemisphere anthropogenic surface $SO_2$ emissions are shown by line contours in Fig.2. The contours start from 20 to 40 ng $m^{-2}s^{-1}$ with an interval of 10 ng $m^{-2}s^{-1}$. The surface $SO_2$ emissions are small before 1950s and rapidly increase during the PHC period. We modified the sentence and mainly focus on the large surface $SO_2$ emissions and sulphate loading during the PHC period (L403-405): "The growing sulphate loading during the PHC period corresponds with the increase in northern-hemisphere anthropogenic surface $SO_2$ emissions (line contours in Fig.2)".

*l389 why "relative"?*
*Response:* **Done.** We delete the word "relatively".

*l389 The more immediate reason is the higher incoming solar flux at lower latitudes.*
*Response:* **Done.** We add the impact of solar flux in L417-419: "Due to the more rapid oxidation and higher incoming solar flux at lower latitudes, an equatorward shift in SO2 emissions around 1990s result in a more efficient production of sulphate and stronger aerosol forcing.".

*l399 Perhaps this should be more cautiously "approximately linearly". See debate of Stevens (2015, doi: 10.1175/JCLI-D-14-00656.1), Kretzschmar et al. (2017, doi:10.1175/JCLI-D-16-0668.1), Booth et al. (2018, doi:10.1175/JCLI-D-17-0369.1).*
**Response: Done.** We change the wording to "approximately linearly" as suggested in L431.

*l402 What is "this nonlinearity"? Before, linear relationships were described.*

*Response:* **Done.** We rewrite the sentence as follows (L429-435): "In both cases, the approximation of linearity holds less well for UKESM1-0-LL, especially at small sulphate loadings. This reflects the behaviour of HadGEM2, a predecessor of UKESM1 (Wilcox et al., 2015), and is likely to be due to the strong aerosol-cloud albedo effect in these models. The global mean annual mean reff decreases by about 0.7 µm since pre-industrial era, more than twice the magnitude of change seen in the other models (Fig.1b in Wilcox et al., 2015 and Fig.9b in this study)."

*l479 Why "generally"? It is of course a coarse approximation only.*
*Response:* **Done.** We replace "generally parametrized as" to "approximated to" in L508.

*l519 The cloud albedo term may also carry a substantial contribution by adjustments, namely via the adjustments of liquid water path.*
*Response:* The ACI term will be influenced by factors contributing to cloud albedo. Liquid water path changes can often be one of the strong drivers of cloud albedo change and can be driven by changes unrelated to aerosol changes (such as global warming). In this study we consider only the radiative changes between the historical simulations and the parallel simulations where aerosol emissions are fixed (hist-piAer). We cannot hold cloud liquid water path fixed in these experiments. So, whilst liquid water changes will be one of a number of factors influencing our ACI term, these should just be capturing those liquid water path changes that are related to the presence or absence of aerosol emission changes.

*l529 This is not quite true. Delta reff is also influenced by changes in liquid water path. The forcing can be identified when investigating droplet number concentrations.*
*Response:* Agreed. we've tried to make this sentence more explicit (L571-573): "$\Delta r_{eff}$ is generally related to the cloud-optical depth and cloud water path, and $\Delta CLT$ is related to adjustments in cloud cover due to ACI."

---

## Author Response (AR2)

In the following, the text with italicization indicates the Reviewers' comments, and the normal text is our response.

**Replies to Reviewer's comments:**

Thank you very much for the insightful comments during the whole process, which lead to a significant improvement. Here is the item-by-item reply to your latest suggestions.

**Suggestions for technical corrections:**

1. *l109: "van Noije"*

   **Response:** Corrected.

2. *l193: I think the information that the forcing sensitivity is not correlated with the effective forcing is important enough to be mentioned here.*

   **Response: Done.** The sentences are modified in L179-180: "The ERFaer for each ESM except MPI-ESM-1-2-HAM is listed in Table 3. The ERFaer is not correlated with the aerosol-forcing-sensitivity."

3. *l324: Units are required in the Table header.*

   **Response: Done.**